# Deubiquitinating activity of SARS-CoV-2 papain-like protease does not influence virus replication or innate immune responses *in vivo*

**Mariska van Huizen**[1], **Jonna R. Bloeme - ter Horst**[1], **Heidi L. M. de Gruyter**[1], **Paul P. Geurink**[2], **Gerbrand J. van der Heden van Noort**[2], **Robert C. M. Knaap**[1], **Tessa Nelemans**[1], **Natacha S. Ogando**[1], **Anouk A. Leijs**[1], **Nadya Urakova**[1], **Brian L. Mark**[3], **Eric J. Snijder**[1], **Sebenzile K. Myeni**[1☯*], **Marjolein Kikkert**[1☯*]

1 Molecular Virology Laboratory, Leiden University Center of Infectious Diseases (LU-CID), Leiden University Medical Center, Leiden, Netherlands, 2 Department of Cell and Chemical Biology, Division of Chemical Biology and Drug Discovery, Leiden University Medical Centre, Leiden, The Netherlands, 3 Department of Microbiology, University of Manitoba, Winnipeg, Canada

☯ These authors contributed equally to this work.
* s.k.myeni@lumc.nl (SKM); m.kikkert@lumc.nl (MK)

## Abstract

The coronavirus papain-like protease (PLpro) is crucial for viral replicase polyprotein processing. Additionally, PLpro can subvert host defense mechanisms by its deubiquitinating (DUB) and deISGylating activities. To elucidate the role of these activities during SARS-CoV-2 infection, we introduced mutations that disrupt binding of PLpro to ubiquitin or ISG15. We identified several mutations that strongly reduced DUB activity of PLpro, without affecting viral polyprotein processing. In contrast, mutations that abrogated deISGylating activity also hampered viral polyprotein processing and when introduced into the virus these mutants were not viable. SARS-CoV-2 mutants exhibiting reduced DUB activity elicited a stronger interferon response in human lung cells. In a mouse model of severe disease, disruption of PLpro DUB activity did not affect lethality, virus replication, or innate immune responses in the lungs. This suggests that the DUB activity of SARS-CoV-2 PLpro is dispensable for virus replication and does not affect innate immune responses *in vivo*. Interestingly, the DUB mutant of SARS-CoV replicated to slightly lower titers in mice and elicited a diminished immune response early in infection, although lethality was unaffected. We previously showed that a MERS-CoV mutant deficient in DUB and deISGylating activity was strongly attenuated in mice. Here, we demonstrate that the role of PLpro DUB activity during infection can vary considerably between highly pathogenic coronaviruses. Therefore, careful considerations should be taken when developing pan-coronavirus antiviral strategies targeting PLpro.

**Data Availability Statement:** All relevant data are in the manuscript and its Supporting information files.

**Funding:** This work was supported by a graduate program grant from the Netherlands Organization for Scientific Research (NWO) to MvH through the LUMC (NWO grant 022.006.010). For this project, MK has received funding from the European Union's Horizon 2020 research and innovation program under grant agreement No 952373. The funders had no role in study design, data collection and analysis, decision to publish, or preparation of the manuscript.

**Competing interests:** The authors have declared that no competing interests exist.

## Author summary

The COVID-19 pandemic highlighted the need for pan-coronavirus targeting antivirals and vaccines. One of the potential targets is the viral papain-like protease (PLpro). PLpro is required for processing of the viral replicase polyprotein, thus enabling virus replication, but it also contributes to immune evasion by its deubiquitinating and deISGylating activities. Here we show that the deubiquitinating activity of SARS-CoV-2 PLpro does not influence virus replication, innate immune responses or lethality in mice. For SARS-CoV, we found that a mutant virus lacking deubiquitinating activity replicated to slightly lower titers and elicited a diminished immune response. This is in strong contrast to our previous findings for MERS-CoV, where removal of the deubiquitinating and deISGylating activities of PLpro strongly attenuated the virus in mice, completely alleviating lethality, and rendered it a promising live-attenuated vaccine candidate. This study uncovers important differences between highly pathogenic coronaviruses in the differential roles of PLpro during infection. These mechanistic insights provide valuable information for the development of pan-coronavirus targeting antivirals and vaccines.

## Introduction

The coronavirus disease 2019 (COVID-19) pandemic highlighted the pandemic potential and global impact that emerging coronaviruses can have [1,2]. Besides severe acute respiratory syndrome coronavirus 2 (SARS-CoV-2), the etiological agent of the COVID-19 pandemic, six other coronavirus species are known to infect humans, including the highly pathogenic SARS-CoV and Middle East respiratory syndrome (MERS)-CoV [3]. Coronaviruses have a plus-stranded RNA genome of 26–32 kilobases [3–5]. Two-thirds of the genome is occupied by two overlapping open-reading frames, ORF1a and ORF1ab. These are translated into two large replicase polyproteins that incorporate up to 16 non-structural proteins (nsps) required for viral RNA replication. The nsps are released from the polyproteins by two internally encoded protease domains, the papain-like protease (PLpro) in nsp3 and the main protease (Mpro) in nsp5. The remaining one-third of the genome encodes four structural proteins that assemble into new virions together with the viral RNA, and additionally this part encodes several accessory proteins.

Coronaviruses manipulate the immune response through several viral proteins [6,7]. Among these is the PLpro domain of nsp3, which is essential for releasing nsp1, nsp2, and nsp3 from the viral replicase polyproteins [8]. Furthermore, it can cleave cellular proteins, such as ULK1 and IRF3 [9,10]. In addition to cleaving polypeptide backbones, SARS-CoV-2 PLpro can also remove ubiquitin and the ubiquitin-like molecule ISG15 from substrate proteins by its deubiquitinating (DUB) and deISGylating activity, respectively. This feature is shared with the PLpro domains encoded by SARS-CoV, MERS-CoV, and other coronaviruses [11–16].

Ubiquitin is an 8-kDa protein that can be conjugated covalently mainly to lysine residues of substrate proteins by a cascade of E1, E2, and E3 enzymes, a process that can be reversed by DUBs [17]. Since ubiquitin contains seven lysine residues (K6, K11, K27, K29, K33, K48, and K63), it can form polyubiquitin chains by building onto these or onto the N-terminal methionine (M1). Different ubiquitin linkages direct diverse outcomes, such as proteasome-mediated degradation of the ubiquitinated target, regulation of protein-protein interactions, and activation of a plethora of cell signaling pathways including the immune response [18,19]. It was shown that the *in vitro* DUB activity of SARS-CoV-2 PLpro is highly specific for K48-linked ubiquitin chains with negligible activity towards other ubiquitin linkages and mono-ubiquitin

[14–16]. K48-linked ubiquitin is the most abundant ubiquitin linkage type in the cell and it is crucial for targeting substrates for proteasome-mediated degradation, thus maintaining proteostasis [20,21]. Furthermore, PLpro can antagonize the IFN-I response by deubiquitinating stimulator of interferon genes (STING) [22].

ISG15 consists of two ubiquitin-like (Ubl) moieties and its expression depends on IFN-I signaling [23,24]. ISG15 can be secreted as a cytokine or it can be conjugated to substrate proteins in a process similar to ubiquitination [23,25]. ISGylation positively regulates innate immune signaling, which is in part mediated through ISGylation of the transcription factor IRF3 [26]. Therefore, also the deISGylating activity of PLpro may interfere with the activation of the immune response. The potential importance of this mechanism is underlined by a study showing that SARS-CoV-2 utilizes its PLpro domain to remove ISG15 from the double-stranded RNA sensor melanoma differentiation-associated protein 5 (MDA5), thereby suppressing the immune response [27]. Furthermore, ISG15 was shown to induce a pro-inflammatory phenotype in SARS-CoV-2-infected macrophages [28]. PLpro counteracts this response by removing ISG15 from various substrates, including glycolytic enzymes that drive pro-inflammatory macrophage polarization [28].

To specifically investigate the role of the DUB and deISGylating activities of PLpro in the context of coronavirus infection, one needs to decouple these activities from the polyprotein processing activity which is essential for virus replication. Previously, using structure-guided mutagenesis, several amino acids were identified that are crucial for specific interactions between SARS-CoV or MERS-CoV PLpro and ubiquitin or ISG15 [29–35]. Mutations that reduced DUB activity also diminished the ability of PLpro to antagonize IFNβ and NFκB activation in reporter assays or cell-based expression systems [29–31]. Infection of immune-competent cells with a SARS-CoV mutant exhibiting reduced DUB activity (hereafter called DUB mutant) yielded lower virus titers and induced increased IFNβ expression [34]. Moreover, our laboratory has recently shown that the DUB and deISGylase mutant of MERS-CoV elicits a stronger or better regulated innate immune response *in vitro* and *in vivo*, respectively [36]. In a mouse model, this mutant virus was strongly attenuated while still replicating robustly in the lungs of infected mice. Upon recovery from infection with DUB and deISGylase mutant MERS-CoV, mice could efficiently clear wild-type MERS-CoV when subsequently challenged with an otherwise lethal virus dose, without developing detectable disease or infection. This suggests that disrupting the DUB and deISGylating activity of MERS-CoV PLpro may provide a strategy for the development of a live-attenuated vaccine [36]. The mutations subsequently described to disrupt the *in vitro* binding of ubiquitin and ISG15 to SARS-CoV-2 PLpro are very comparable to those described for SARS-CoV, which is not surprising given the strong sequence similarity between the PLpro domains of the two viruses [15,16,37]. However, the impact of these mutations on SARS-CoV-2 replication and the elicited antiviral responses have not been described thus far.

To elucidate the role of the DUB and deISGylating activities of SARS-CoV-2 PLpro during virus infection, we generated PLpro mutant viruses in which these activities were disrupted. We studied these mutant viruses *in vitro* and *in vivo* and reveal important differences between SARS-CoV, SARS-CoV-2, and MERS-CoV in terms of the contribution of the PLpro DUB activity to virus replication, immune evasion, and disease outcome.

## Results

### Structure-guided perturbation of the binding of ubiquitin and ISG15 to SARS-CoV-2 PLpro

SARS-CoV and SARS-CoV-2 PLpro have a high degree of similarity at the amino acid sequence level (83% identity), in their three-dimensional structure, and in their substrate

binding properties [8,15,16,37]. This can be exploited to gain insight into the binding of K48-linked diubiquitin to SARS-CoV-2 PLpro. Since a crystal structure of K48-linked diubiquitin in complex with SARS-CoV-2 PLpro was not available when we were engineering mutations to disrupt its DUB activity, we superimposed the crystal structures of SARS-CoV PLpro in complex with K48-linked diubiquitin (PDB 5E6J [31]) and SARS-CoV-2 PLpro in complex with the C-terminal domain of ISG15 (PDB 6XA9 [15]) (Fig 1A). As ISG15 and ubiquitin interact differently with SARS-CoV-2 PLpro [38], this offered a structural basis for the generation of mutants exhibiting a specific loss of either the DUB or the deISGylating activity, without disrupting viral polyprotein processing (Figs 1A, 1B, and S1A). This could be accomplished through mutating PLpro residue M208 located in the S1 site of the fingers subdomain. M208 interacts with the hydrophobic I44 patch of the proximal $^{S1}$Ub moiety, but not with ISG15 (Fig 1C) [31]. The introduction of a serine at this position was predicted to result in the disruption of hydrophobic interactions (S1-Ub mutant; Fig 1C and 1D). Furthermore, PLpro residues F69 and H73, which are located in the S2 site of the thumb subdomain, interact with the hydrophobic I44 patch of the distal $^{S2}$Ub moiety (Fig 1E). As this structure did not include the N-terminal domain of ISG15, the interaction between F69 and H73 and human ISG15 could not be evaluated. A previous study described interactions between PLpro residues V66 and F69 and mouse ISG15 residues A2, T20, and M23 (S1B Fig) [16]. However, A2 and T20 are not conserved in human ISG15 (S1D Fig). The introduction of residues with different size or side chain polarity at these positions in PLpro could influence binding to both ubiquitin and ISG15 (S2-Ub-mutant; Figs 1F and S1C). Additionally, PLpro residues R166 and E167 were predicted to be crucial for the interaction with both ubiquitin and ISG15. R166 forms a hydrogen bond with Q49 in $^{S1}$Ub, while E167 forms hydrogen bonds with R42 in $^{S1}$Ub and R153 in ISG15 (S1-Ub/ISG15 mutant; Fig 1G and 1H). These interactions can potentially be perturbed by the introduction of uncharged or oppositely charged residues (Fig 1I and 1J). Previously, it was suggested that mutating W107 and A108 in SARS-CoV PLpro (W106 and A107 in SARS-CoV-2 PLpro), which are adjacent to the exit tunnel of the enzyme's active site, could specifically reduce the deISGylating activity, while leaving DUB activity intact (Fig 1K) [31]. Therefore, we also included these positions in our mutagenesis study (S1'-ISG15 mutant).

## Biochemical characterization of SARS-CoV-2 PLpro mutants

To evaluate the impact of the designed mutations or combinations of mutations on the DUB and deISGylating activities of SARS-CoV-2 PLpro, we expressed various PLpro mutant proteins in *Escherichia coli* and purified them. To characterize the enzyme kinetics, recombinant PLpro mutant proteins were incubated with three fluorogenic 7-amido-4-methylcoumarin (AMC) substrates, Ub-AMC, K48-linked diubiquitin-AMC (K48-diUb-AMC), and ISG15-AMC, and AMC release was measured over time. The S2-Ub mutants F69S, F69R, and F69S/E70K/H73G (FEH) hydrolyzed Ub-AMC at similar rates as wild-type PLpro, which was expected since Ub-AMC does not interact with the S2 site (Fig 2A and 2D and S1 Table). Hydrolysis of K48-diUb-AMC by these mutants was strongly reduced, indicating that these mutations disrupted DUB activity (Fig 2B and 2D and S1 Table), which is also in line with previously published data [15,37]. All other mutants displayed reduced activity towards both Ub-AMC and K48-diUb-AMC (Fig 2A, 2B and 2D). Like the DUB activity, deISGylating activity towards ISG15-AMC was completely abrogated for PLpro W106L/A107S (S1'-ISG15 mutant) and R166S/E167R (S1-Ub/ISG15-mutant; Fig 2C and 2D). Furthermore, PLpro M208R (S1-Ub mutant) and FEH (S2-Ub mutant) displayed reduced deISGylating activity. Interestingly, PLpro FEHM (S1+S2-Ub mutant) which is a combination of FEH and M208S, did not

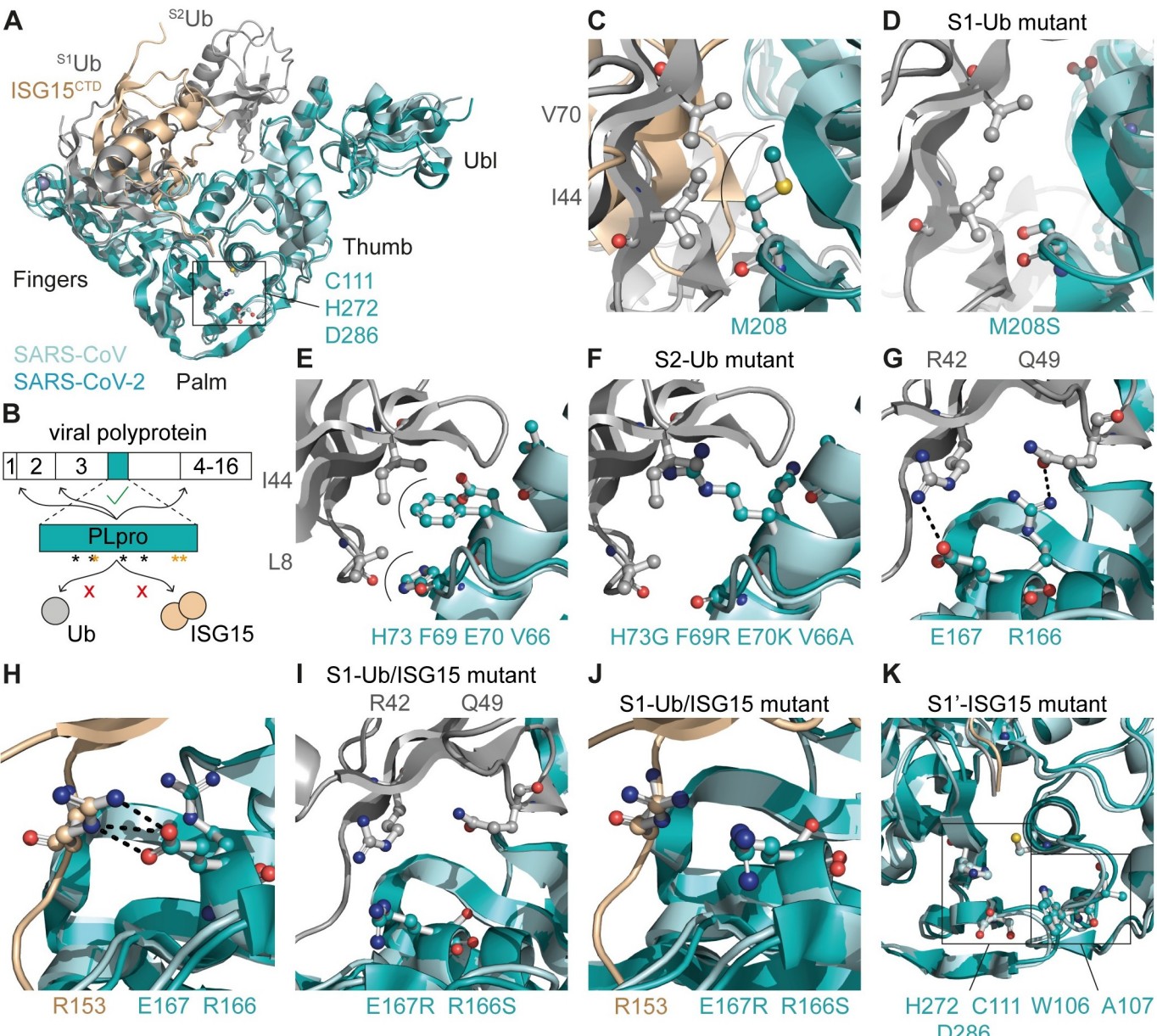

**Fig 1. Perturbation of the binding of SARS-CoV-2 PLpro to ubiquitin and ISG15.** (A) Cartoon representation of SARS-CoV PLpro (aquamarine) in complex with K48-linked diubiquitin (grey; PDB 5E6J) aligned with SARS-CoV-2 PLpro (teal) in complex with ISG15CTD (wheat; PDB 6XA9). Catalytic triad residues (C111, H272, D286) are shown in ball-and-stick representation. (B) Schematic representation of the localization of the PLpro domain in the viral polyprotein. PLpro is responsible for cleaving the junctions between nsp1, nsp2, nsp3, and nsp4. Black asterisks indicate positions where mutations were introduced to disrupt the binding to ubiquitin or ISG15, without affecting viral polyprotein processing. Orange asterisks indicate catalytic residues. (C) M208 of PLpro interacts with I44 and V70 in the hydrophobic patch of S1ubiquitin. (D) Introduction of M208S is predicted to disrupt the interaction with ubiquitin. (E) Interaction of PLpro residues V66, F69, E70, and H73 with S2ubiquitin residues L8 and I44. (F) Introduction of V66A, F69R, E70K, and H73G is predicted to disrupt the interaction with S2ubiquitin. (G-H) PLpro residues R166 and E167 form hydrogen bonds with S1ubiquitin residues Q49 and R42, respectively (G) and ISG15 residue R153 (H). (I-J) Introduction of R166S and E167R is predicted to disrupt the hydrogen bonds between PLpro and S1ubiquitin (I) and ISG15 (J). (K) Localization of W106 and A107 in the S1' site, in close proximity to the catalytic triad residues C111, H272, and D286. (C-K) Residues that mediate interaction are shown in ball-and-stick representation. Hydrophobic interactions are indicated by arches and hydrogen bonds by black dashed lines.

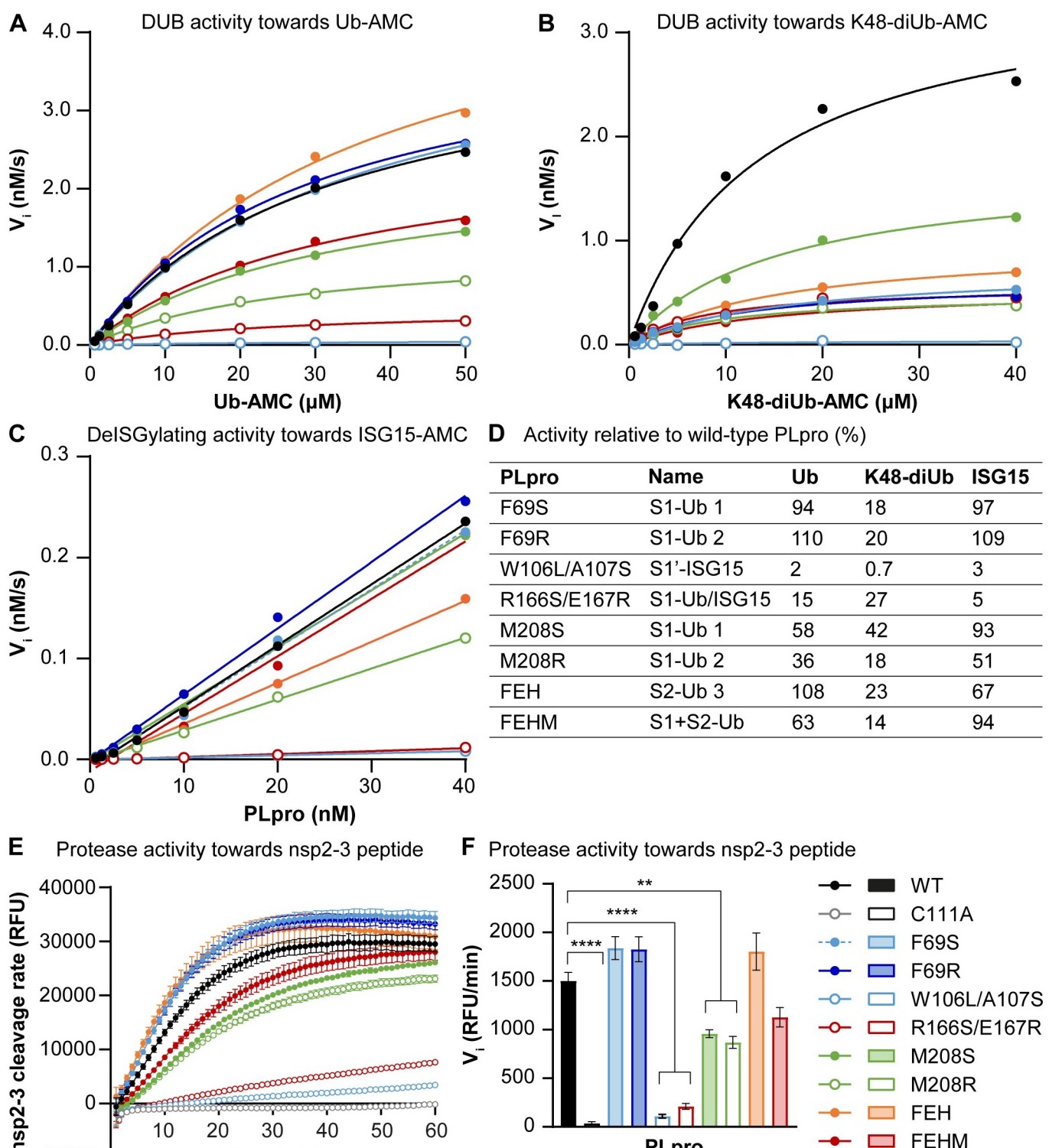

**Fig 2. DUB, deISGylating and polyprotein processing activity of SARS-CoV-2 PLpro mutants.** (A-B) Michaelis-Menten kinetics comparing hydrolysis of a range of Ub-AMC (A) and K48-diUb-AMC (B) concentrations by SARS-CoV-2 wild-type PLpro and mutants. (C) Kinetics of a range of SARS-CoV-2 PLpro concentrations towards ISG15-AMC. (D) Activity of PLpro mutants on Ub-AMC, K48-diUb-AMC, and ISG15-AMC relative to wild-type PLpro, as calculated from the $k_{cat}/K_M$ values for Ub-AMC and K48-diUb-AMC. For ISG15-AMC, the slope of the curves plotted in panel C were used to calculate relative activity. Naming of mutants corresponds to Fig 1. (E) Polyprotein processing activity of SARS-CoV-2 wild-type PLpro and mutants as assessed by cleavage of a FRET

peptide corresponding to the nsp2-3 junction. (F) Initial velocities were extrapolated from the linear portion of the curve as a measure for the substrate cleavage rate. FEH(M) = F69S/E70K/H73G(/M208S) Representative examples are shown of one to three experiments each performed in technical triplicate (A-D). Data in E-F is represented as mean ± s.e.m. of 3 experiments and was analyzed by one-way ANOVA with Dunnett's multiple comparisons test, comparing each group to wild-type PLpro. * $p < 0.05$, ** $p < 0.01$, *** $p < 0.001$, **** $p < 0.0001$.

show reduced deISGylating activity. The S2-Ub mutants F69S and F69R and the S1-Ub mutant M208S hydrolyzed ISG15-AMC at a similar rate as wild-type PLpro.

To assess the DUB activity in a cellular context, we introduced the mutations into a sequence encoding V5-tagged PLpro expressed from a mammalian expression vector. PLpro mutants, and wild-type and catalytically inactive (C111A) control proteins, were expressed in HEK293T cells together with HA-tagged ubiquitin. Total levels of conjugated HA-tagged ubiquitin were analyzed by western blot. Wild-type PLpro reduced ubiquitin conjugation by its DUB activity (S2A Fig). Several mutations resulted in decreased DUB activity, including W106L and W106L/A107S in the S1' site, R166S/E167R in the S1 site, and F69S, F69R, and FEH in the S2 site, which is in line with the kinetic measurements.

To also assess the deISGylating activity in a cellular context, we co-expressed ISG15, along with the enzymes needed for ISG15 conjugation (Ube1L, UbcH8, and Herc5), and the PLpro mutants in HEK293T cells. Wild-type PLpro by its deISGylating activity strongly reduced ISG15 conjugation (S2B Fig). Of all the mutants that were tested, only PLpro E167R and R166S/E167R showed profoundly reduced deISGylating activity. It should be noted that the expression levels of those mutant PLpro proteins were also lower, which was observed consistently across all assays. This suggests that the mutations influence the expression level and/or turn-over of these specific mutant PLpro proteins.

To create viable virus mutants, it is essential that viral polyprotein processing remains intact. Therefore, we investigated whether the mutations had any effect on viral polyprotein processing. To this end, a FRET peptide was designed based on the nsp2-3 junction, FTLKGG|APTKVT. Upon cleavage of the peptide by PLpro, quenching of the EDANS signal by DABCYL is disrupted leading to a fluorescent signal emitted by EDANS. The DUB mutants F69S, F69R, FEH, and F69S/E70K/H73G/M208S (FEHM) all cleaved the nsp2-3 peptide at a rate comparable to wild-type PLpro (Fig 2E and 2F). PLpro M208S and M208R had a 2-fold reduced activity. The DUB- and deISGylase-deficient mutants W106L/A107S and R166S/E167R had almost completely lost nsp2-3 cleavage activity, indicating that these mutants have a more general defect in the protease activity. These mutations are located close to the catalytic core of PLpro and may thereby cause structural changes that also impact recognition and cleavage of the viral polyprotein. Disruption of the deISGylating activity thus appeared rather challenging, since the two mutants that displayed strongly reduced DUB and deISGylating activity were also hampered in viral polyprotein processing.

Altogether, we identified six mutants (F69S/R, FEH, FEHM, and M208S/R) with impaired DUB activity towards K48-linked diUb for which processing of the viral polyprotein-derived peptide is intact or only mildly reduced.

## SARS-CoV-2 DUB mutants display unaltered growth kinetics in human lung cells

Having identified (combinations of) mutations that reduce the DUB and deISGylating activity of SARS-CoV-2 PLpro, we used a full-length SARS-CoV-2 cDNA clone to introduce them into the viral genome as to assess the impact of the mutations on virus replication. *In vitro*-generated viral RNA was launched through electroporation of BHK-21 cells, which were

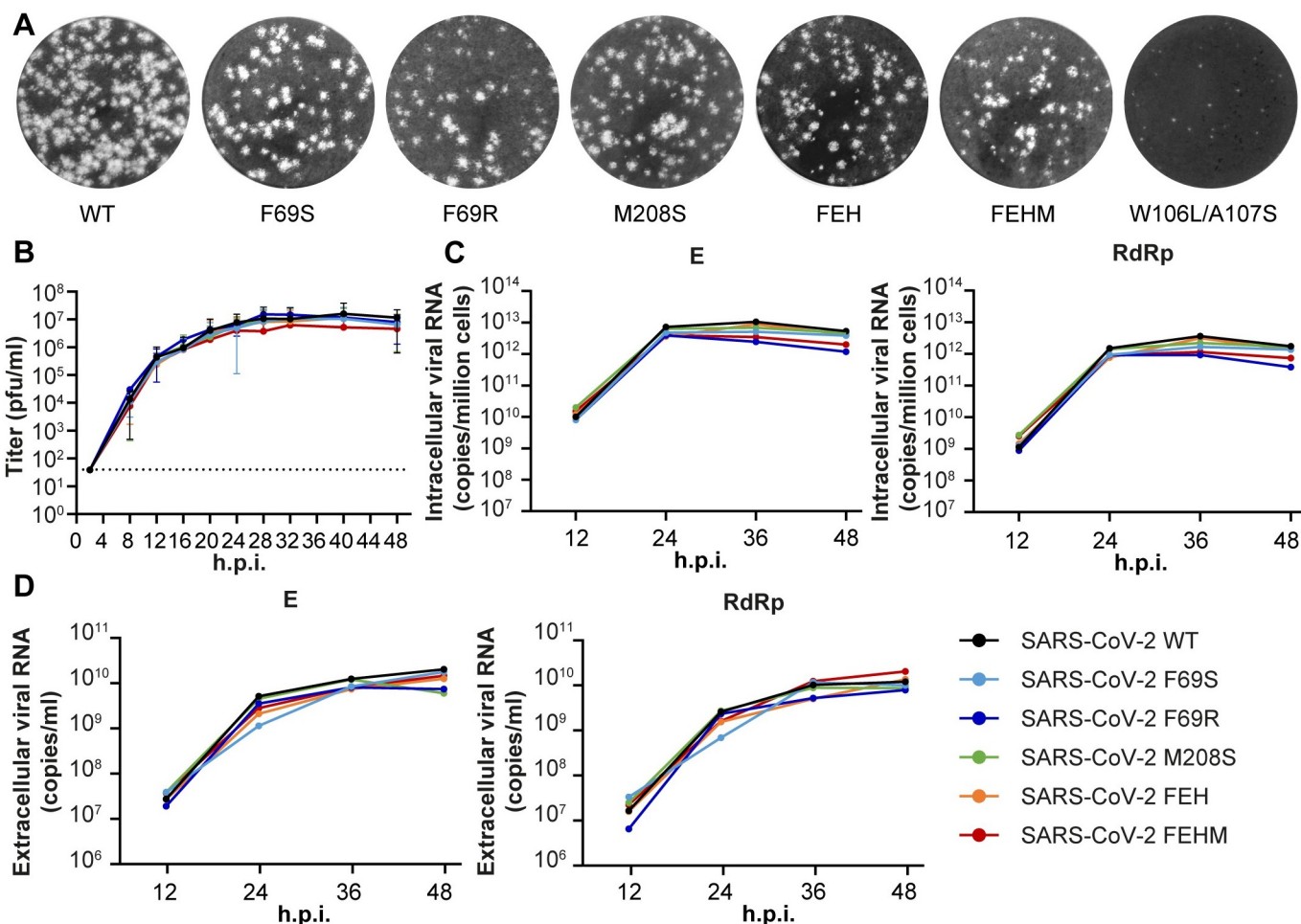

**Fig 3. SARS-CoV-2 DUB mutant viruses display unaltered growth kinetics in human lung cells.** (A) SARS-CoV-2 stock titers were determined by plaque assay on VeroE6 cells. Representative examples of the plaque morphology of wild-type and PLpro mutant SARS-CoV-2 are shown. (B) Calu-3 cells were infected at MOI 0.1 with wild-type or DUB mutant SARS-CoV-2 to determine the replication kinetics. Virus titers in the supernatant were determined by plaque assay with a limit of detection of 40 plaque-forming units (pfu)/ml (indicated by the dotted line). At 2 hpi, inoculum was removed and cells were washed three times. The last wash was collected and taken along in the plaque assay. No plaques were detected, meaning that the amount of residual virus was below 40 pfu/ml. (n = 5). Data represented as mean ± s.e.m. (C-D) Intracellular (C) and extracellular (D) viral RNA levels were determined by RT-qPCR using primers targeting the E and RdRp coding regions (n = 3). Representative examples are shown. FEH(M) = F69S/E70K/H73G(/M208S). Data was analyzed by one-way ANOVA with Dunnett's multiple comparisons test, comparing each group to wild-type SARS-CoV-2. No significant differences were found.

subsequently mixed 1:1 with SARS-CoV-2-susceptible VeroE6 cells. Progeny viruses were then passaged twice in immune-competent Calu-3 cells and the presence of the engineered mutations as well as the absence of any unintended changes was confirmed by Sanger sequencing. Five out of seven virus mutants had a plaque morphology comparable to that of the wild-type virus (Fig 3A). The S1' site mutant W106L/A107S on the other hand yielded very small plaques and the initial titers in the medium of the electroporated cells were three orders of magnitude lower than those of the wild-type virus, indicating that this mutant is severely crippled. The S1 site mutant R166S/E167R, which exhibited both reduced DUB and deISGylating activity, did not yield any viable progeny. This is most likely due to a defect in viral replicase polyprotein processing since these mutants also displayed a strong reduction in cleavage of the nsp2-3 peptide.

We determined the replication kinetics of the remaining five DUB mutant viruses in Calu-3 cells. Virus titers in supernatants harvested at 4-hour intervals between 8 and 48 hours post infection (hpi) were determined by plaque assay (Fig 3B). In addition, intracellular and extra-cellular viral RNA levels were measured by real-time quantitative PCR (RT-qPCR) (Fig 3C and 3D). There were no significant differences between the DUB mutants and the wild-type control in terms of either progeny virus titers or viral RNA levels, suggesting that these mutations at the PLpro ubiquitin-binding surface do not interfere with virus replication.

## SARS-CoV-2 DUB mutants elicit an increased IFNβ response early in infection of human lung cells

To evaluate the potential role of PLpro DUB activity in SARS-CoV-2 immune evasion, we measured expression of *IFNB* and several ISGs and cytokines in infected Calu-3 cells by RT-qPCR. Throughout the course of wild-type SARS-CoV-2 infection, the expression of all genes analyzed was induced compared to mock-infected cells (Fig 4A). At 48 hpi, expression was decreasing again, except for *ISG15*. Compared to the wild-type virus, several DUB mutants elicit increased expression of inflammatory genes. At 12 hpi, SARS-CoV-2 PLpro mutant F69R induced higher expression of *IFNB*, *IFIT2*, *RSAD2*, and *IL6* compared to the wild-type virus (Fig 4B). SARS-CoV-2 M208S induced increased expression of *IFIT2*, while SARS-CoV-2 FEHM induced higher *RSAD2* expression at 12 hpi. At 24 hpi, SARS-CoV F69S and M208S induced more *IFNB*, *ISG15*, *IFIT2*, and *RSAD2* expression compared to the wild-type virus (Fig 4C). Additionally, SARS-CoV-2 FEHM induced higher expression of *IFIT2* and *RSAD2*. At 48 hpi, there was no longer a significant difference in gene expression between cells infected with the wild-type or DUB mutant viruses (Fig 4D). We also analyzed the protein levels of ISG15 at 24 hpi by western blot. ISG15 was more abundant in the infected cells compared to the mock (S3 Fig). Furthermore, the DUB mutant viruses induced higher ISG15 protein levels than the wild-type virus, which was in line with the increased mRNA levels at 12 and 24 hpi. In conclusion, these results indicate that disrupting the DUB activity of PLpro impairs the ability of the virus to suppress the innate immune response early after infection.

## The DUB activity of SARS-CoV-2 PLpro is dispensable for virus replication and lethality *in vivo*

Three of the SARS-CoV-2 DUB mutants that were less capable of suppressing the IFN-I response were selected for phenotypic characterization *in vivo*. To this end, we used K18-hACE2 mice, which are a model for severe SARS-CoV and SARS-CoV-2 infection and develop a dose-dependent lung disease with similar clinical features to severe human COVID-19 and development of ARDS, making this model suitable for studying pathogenesis [39–44]. Mice were inoculated intranasally with a lethal dose of $1x10^4$ plaque forming units (pfu) of wild-type SARS-CoV-2 or DUB mutants F69S, F69R, or M208S. Bodyweight and clinical discomfort were monitored daily. Mice were sacrificed upon reaching their humane endpoint, which was defined as a 20% weight loss and/or severe clinical discomfort. Mice from all groups except the mock-infected controls reached humane endpoints and were sacrificed between days 5 and 7 post infection (dpi), while each group also had one to three mice that recovered from the infection (Fig 5A). Minor differences in loss of bodyweight were observed (Fig 5B), but these did not translate into significant lethality differences between the groups infected with wild-type or DUB mutant viruses. In a similar experiment, mice were sacrificed at 2 and 4 dpi and lungs were collected. Virus titers in the lungs were determined by plaque assay (Fig 5C) and viral RNA levels were determined by RT-qPCR (Fig 5D). There was no difference in virus titer or viral RNA levels in the lungs of mice infected with any of the mutant viruses

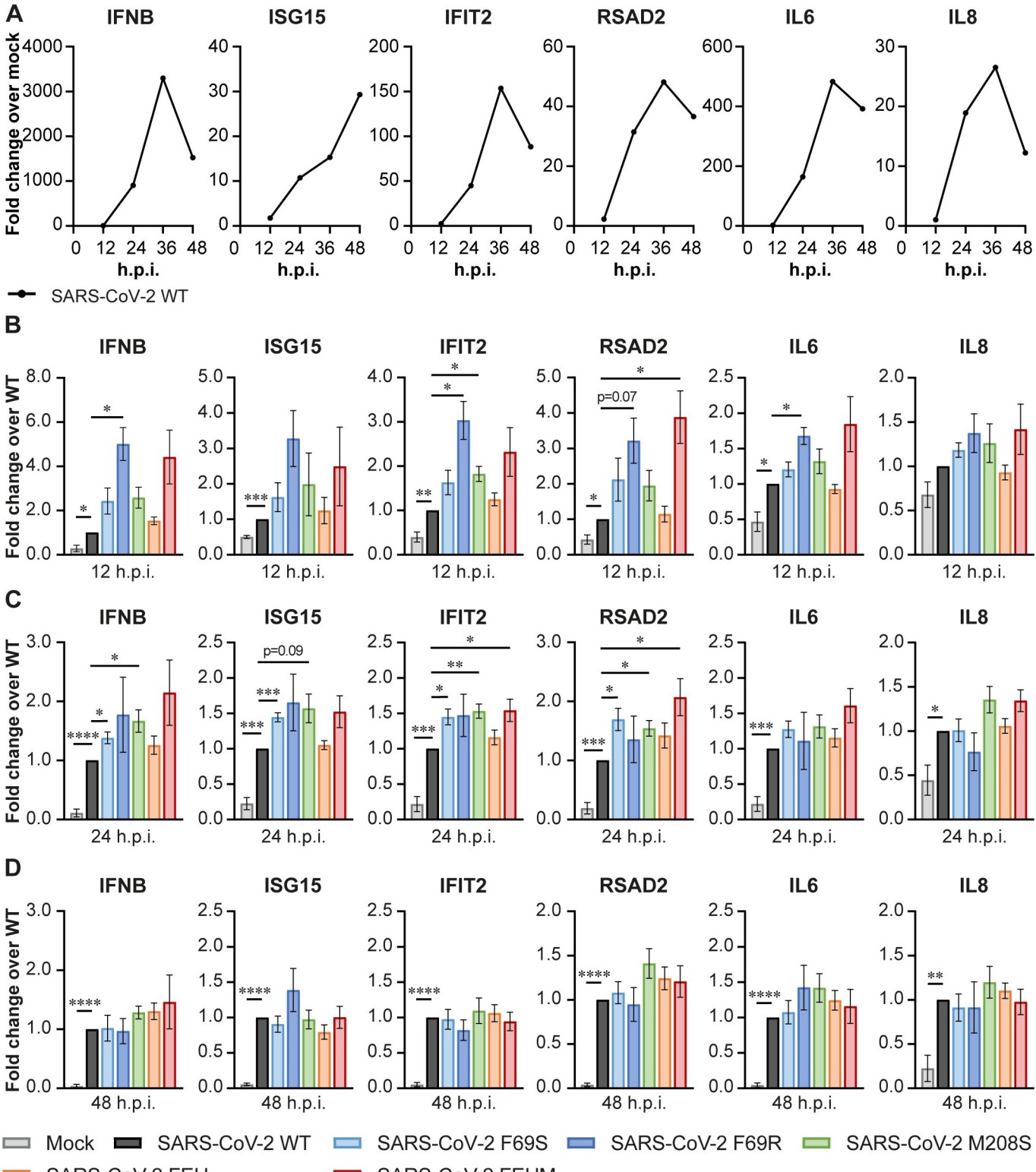

**Fig 4. SARS-CoV-2 DUB mutant viruses elicit a stronger immune response in Calu-3 cells.** Calu-3 cells were infected at MOI 0.1 with wild-type or DUB mutant SARS-CoV-2 to analyze the expression of genes related to the immune response by RT-qPCR. (A) Gene expression over time in Calu-3 cells infected with wild-type SARS-CoV-2 as compared to mock-infected cells. (B-D) Expression of genes related to the immune response at 12 (B), 24 (C), and 48 (D) hours post infection with wild-type or DUB mutant SARS-CoV-2. FEH(M) = F69S/E70K/H73G(/M208S). Panel A shows representative examples. Data in panels B-D are

represented as mean ± s.e.m. of 5–8 independent experiments. Data was analyzed by mixed-effects analysis with Dunnett's multiple comparisons test, comparing each group to wild-type SARS-CoV-2. * p<0.05, ** p<0.01, *** p<0.001, **** p<0.0001.

either at 2 or at 4 dpi. To verify that the mutant viruses were still carrying the engineered PLpro mutations, we amplified the PLpro region by RT-PCR on viral RNA isolated from the lungs of infected mice at 4 dpi and confirmed the presence of the mutations by Sanger sequencing (S4 Fig). Much to our surprise, we conclude that disrupting the DUB activity of SARS-CoV-2 PLpro does not affect virus replication or lethality in this mouse model.

## Disrupting SARS-CoV-2 DUB activity does not affect immune responses measured in the lungs of infected mice

Although no differences in lethality or virus replication were observed, we questioned whether the immune responses in the lungs of mice differed between wild-type and DUB mutant virus infection, as we had observed differences in human lung cells (Fig 4). Therefore, we measured expression of *IFNB*, *IFNL* and several ISGs and cytokines by RT-qPCR. At 2 dpi, there was quite some variation in the immune response within the infected groups, resulting in large

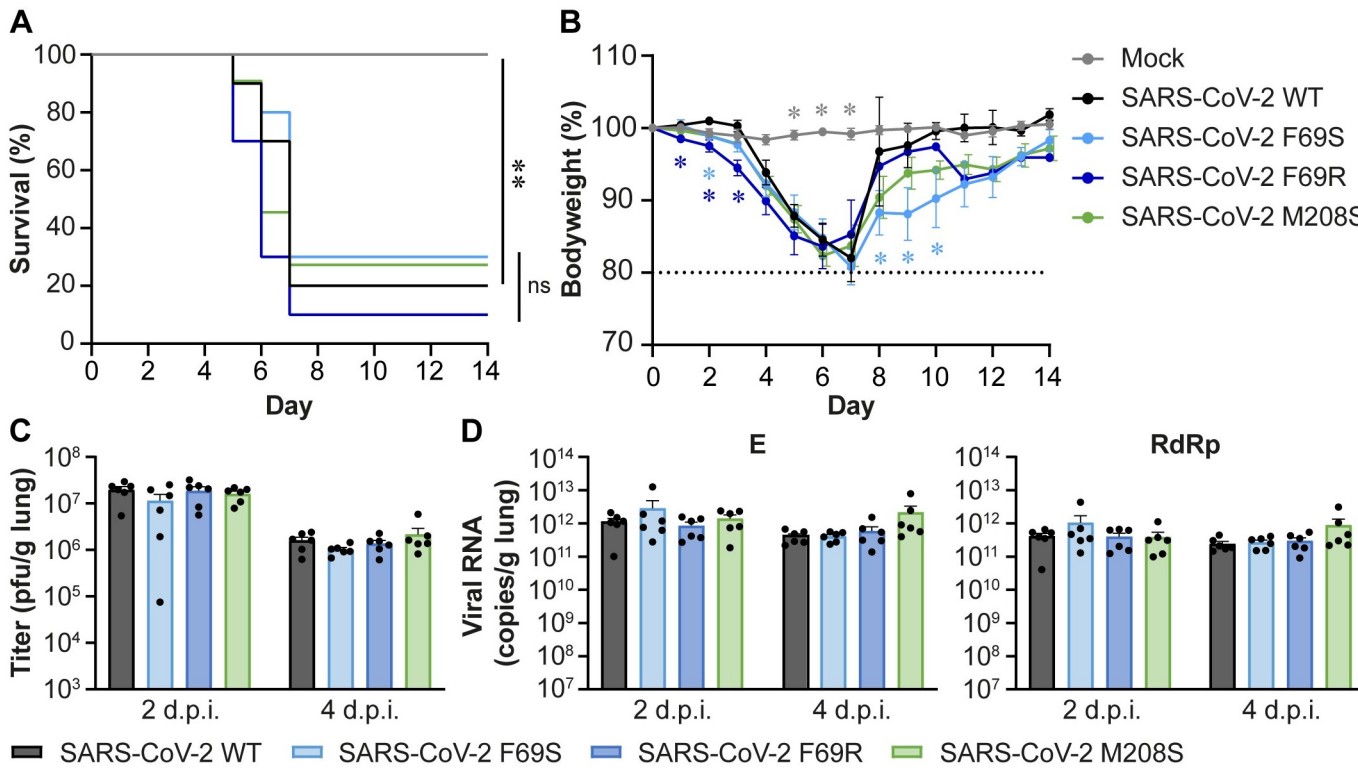

**Fig 5. Disrupting the DUB activity of SARS-CoV-2 PLpro does not affect virus replication or disease in K18-hACE2 mice.** K18-hACE2 mice were infected intranasally with $1 \times 10^4$ pfu wild-type or DUB mutant SARS-CoV-2 or mock-infected with DMEM. (A) Survival curves (%). (B) Bodyweight loss (% from initial weight). Dashed line indicates 20% weight loss upon which mice were euthanized. (C) Virus titers in the lungs at 2 or 4 dpi. Virus titers were determined by plaque assay on VeroE6 cells. (D) Viral genomic and subgenomic RNA levels in the lungs at 2 and 4 dpi. Viral RNA levels were determined by RT-qPCR using primers targeting the E (genomic and subgenomic RNA) and RdRp (genomic RNA) coding regions. n = 8 mice per group for the mock; n = 10 mice per group for SARS-CoV-2 wild-type, F69S, and F69R; n = 11 mice per group for SARS-CoV-2 M208S (A-B). n = 6 mice per group (C-D). Data was analyzed using log-rank test (A) and one-way ANOVA with Dunnett's multiple comparisons test, comparing each group to wild-type SARS-CoV-2 (B-D). ns: not significant, * p<0.05, ** p<0.01, *** p<0.001, **** p<0.0001.

standard deviations (Fig 6A). Despite this variation, expression of *IFNB*, *ISG15*, and *IL6* was significantly induced in the infected groups relative to the mock-infected group. *IFIT2*, *CCL5*, and *TNF* expression were also increased but with less convincing p-values. At 4 dpi, expression of all genes measured, except for *CXCL10*, was significantly induced in the infected groups when compared to the mock-infected control (Fig 6B), although in general induction was lower than at 2 dpi. We did not find clear differences in the induction of these genes between the DUB mutants and wild-type SARS-CoV-2. Our analysis of the expression of a set of highly relevant innate immune genes thus shows that the DUB activity of SARS-CoV-2 PLpro is dispensable for viral innate immune evasion *in vivo*.

## Disrupting the DUB activity of SARS-CoV PLpro reduces virus replication *in vivo*

Considering that SARS-CoV and SARS-CoV-2 PLpro are closely related, we investigated whether DUB activity is also dispensable for SARS-CoV replication and immune evasion. Previously published data on the biochemical characterization of SARS-CoV PLpro showed that introduction of F70S (equivalent to F69S in SARS-CoV-2 PLpro) or M209S (equivalent to M208 in SARS-CoV-2 PLpro) into SARS-CoV PLpro reduced its DUB activity (S5A Fig). Another study showed that introduction of M209R into SARS-CoV PLpro impairs the ability of the virus to suppress IFNβ signaling in Calu-3 cells [34]. Interestingly, the SARS-CoV DUB mutants F70S and M209S were less capable of inhibiting IFNβ promoter activity in a luciferase reporter assay, while the equivalent mutations had no effect on the ability of SARS-CoV-2 PLpro to inhibit IFNβ promotor activity (S5B Fig).

To assess the pathogenesis of SARS-CoV DUB mutants *in vivo*, we inoculated K18-hACE2 mice intranasally with $1x10^4$ pfu of wild-type SARS-CoV or PLpro mutants F70S or M209S and monitored bodyweight and clinical discomfort daily. Mice were sacrificed upon reaching the humane endpoint, which was defined as 20% weight loss and/or severe clinical discomfort. All mice, except for one animal in the SARS-CoV F70S group, had to be euthanized between days 5 and 9 post infection (Fig 7A). There were no significant differences in survival or loss of bodyweight (Fig 7A and 7B). In a separate experiment, we sacrificed mice at 2 and 4 dpi and collected the lungs. Virus titers in the lungs of infected mice were determined by plaque assay (Fig 7C) and viral RNA levels were determined by RT-qPCR (Fig 7D). At 2 dpi, there was no difference in lung viral titers between animals infected with the wild-type virus or the DUB mutants. However, at 4 dpi, SARS-CoV F70S titers were 2-fold lower and SARS-CoV M209S titers were 2.5-fold lower compared to the wild-type control. Viral RNA levels were 2.5-fold lower at 2 dpi for SARS-CoV F70S. At 4 dpi, they were similar to the wild-type virus. SARS-CoV M209S produced similar viral RNA levels as the wild-type control on both days. Overall, DUB mutant progeny titers were slightly lower than those of wild-type SARS-CoV, even though this did not translate into lethality differences. To make sure that the mutant viruses were still carrying the engineered mutations, we amplified the PLpro region by RT-PCR on viral RNA isolated from the lungs of infected mice at 4 dpi and confirmed the presence of the mutations by Sanger sequencing (S6 Fig). Interestingly, disrupting the DUB activity of SARS-CoV PLpro modestly reduced the levels of infectious virus and viral RNA *in vivo*, which is in contrast to what we found for SARS-CoV-2.

## Reduced immune response towards SARS-CoV PLpro mutant F70S *in vivo*

To determine whether the reduced virus titers for the SARS-CoV DUB mutants coincided with changes in the immune response, we determined the expression level of several interferon-related genes and cytokines in the lungs of infected mice by RT-qPCR. At 2 dpi, there

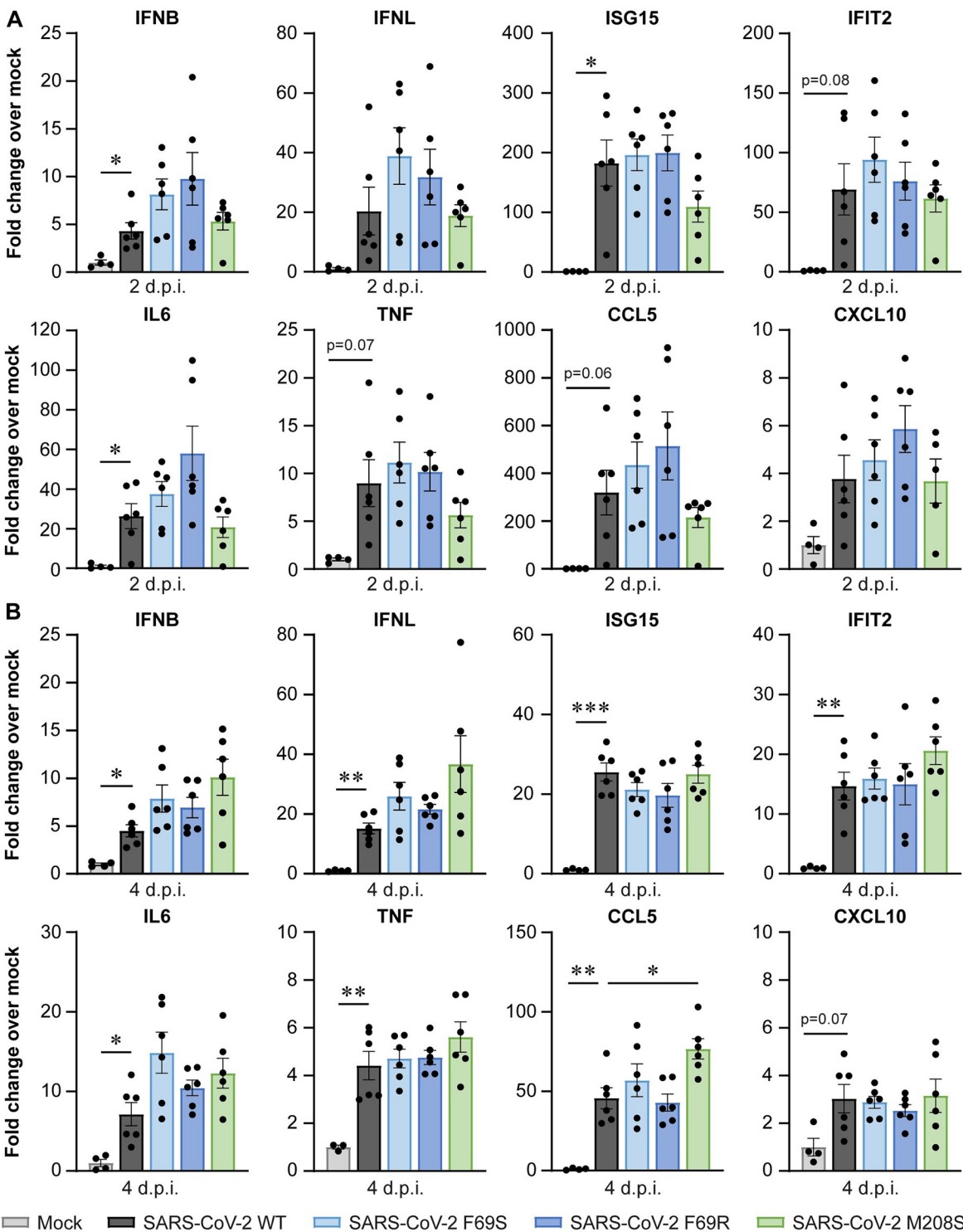

**Fig 6. Disrupting the DUB activity of SARS-CoV-2 PLpro does not affect the immune response in the lungs of infected K18-hACE2 mice.** K18-hACE2 mice were infected intranasally with $1 \times 10^4$ pfu wild-type or DUB mutant SARS-CoV-2 or mock-infected with DMEM. Lungs were collected at 2 and 4 dpi and immune responses were analyzed by RT-qPCR. (A) Immune responses at 2 dpi. (B) Immune responses at 4 dpi. N = 4 mice per group for the mock and n = 6 mice per group for the SARS-CoV-2-infected groups. Data are represented as mean ± s.e.m. Data were analyzed using Welch ANOVA with Dunnett's T3 multiple comparisons test, comparing each group to wild-type SARS-CoV-2. * $p < 0.05$, ** $p < 0.01$, *** $p < 0.001$.

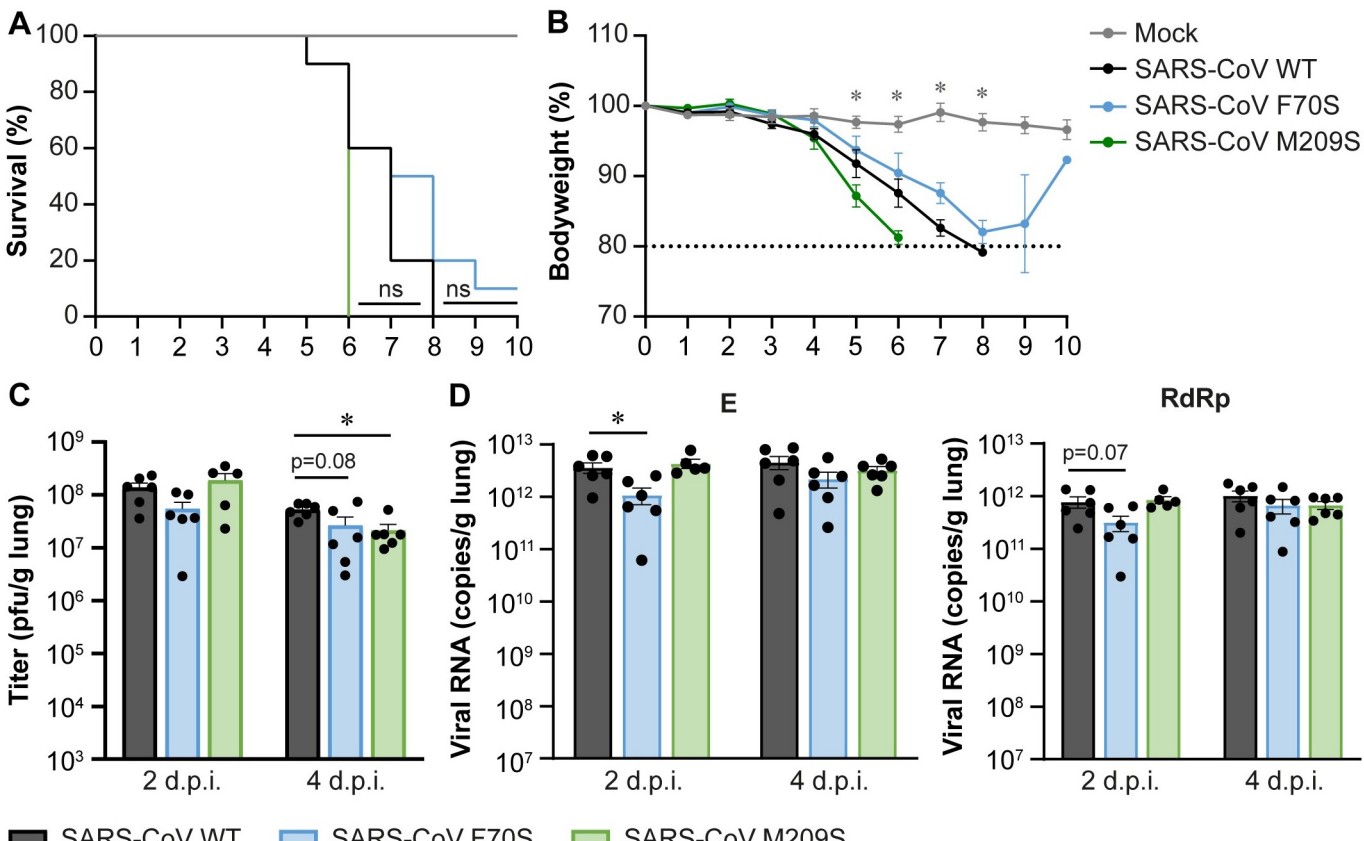

**Fig 7. SARS-CoV DUB mutants replicate to slightly lower titers in K18-hACE2 mice.** K18-hACE2 mice were infected intranasally with $1 \times 10^4$ pfu wild-type or DUB mutant SARS-CoV or mock-infected with DMEM. (A) Survival curves (%). (B) Bodyweight loss (% from initial weight). Dashed line indicates 20% weight loss upon which mice were euthanized. (C) Virus titers in the lungs at 2 or 4 dpi. Virus titers were determined by plaque assay on VeroE6 cells. (D) Viral genomic and subgenomic RNA levels in the lungs at 2 and 4 dpi. Viral RNA levels were determined by RT-qPCR using primers targeting the E (genomic and subgenomic RNA) and RdRp (genomic RNA) coding regions. n = 8 mice per group for mock and n = 10 for SARS-CoV infected groups (A-B). n = 6 mice per group (C-D). Data was analyzed using log-rank test (A) and one-way ANOVA with Dunnett's multiple comparisons test, comparing each group to wild-type SARS-CoV (B-D). *p<0.05, ** p<0.01, *** p<0.001, **** p<0.0001.

was no significant induction of *IFNB*, *IFNL*, *TNF*, and *CCL5* relative to mock-infected animals (Fig 8A). *ISG15* and *IFIT2* were induced ~50-fold in mice infected with wild-type SARS-CoV. Interestingly, expression of these genes was reduced upon infection with SARS-CoV F70S compared to the wild-type control. *IL6* was induced to similar levels by the wild-type virus and DUB mutants. *CXCL10* was induced 90-fold upon wild-type virus infection, and this was reduced in the SARS-CoV F70S-infected group. At 4 dpi, expression of all genes was induced, but there were no differences between animals infected with wild-type SARS-CoV or F70S. SARS-CoV M209S induced similar expression as the wild-type control at both timepoints after infection. We conclude that introduction of F70S into SARS-CoV PLpro diminishes the immune response during the initial phase of the infection in this animal model, which coincides with slightly reduced viral RNA levels (Fig 7D).

## Discussion

Coronavirus proteases are multifunctional enzymes that, in addition to being essential for the maturation of the viral replicase polyproteins, also interfere with host responses that induce an

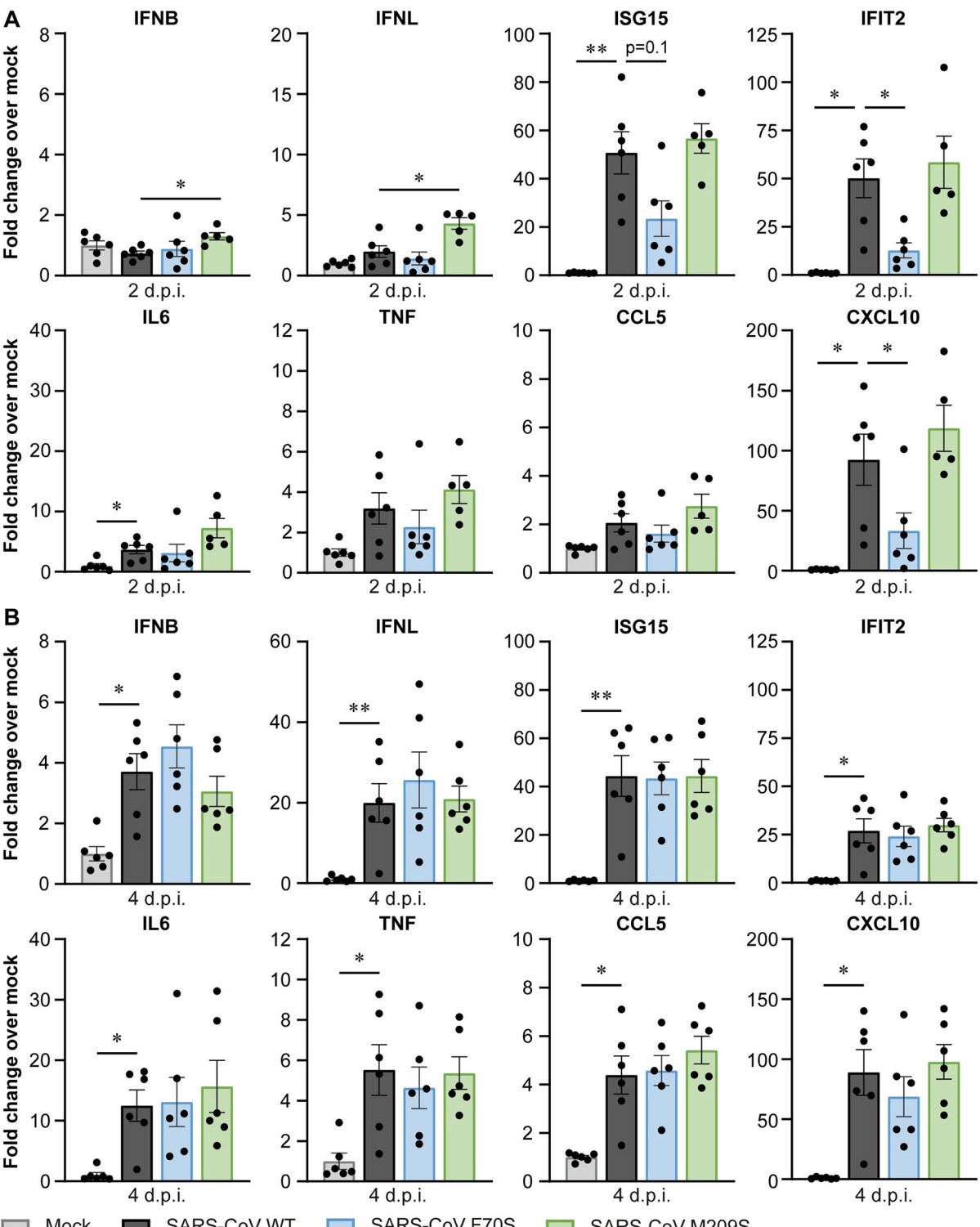

**Fig 8. SARS-CoV F70S elicits a lower immune response in the lungs of infected K18-hACE2 mice.** K18-hACE2 mice were infected intranasally with $1 \times 10^4$ pfu wild-type or DUB mutant SARS-CoV or mock-infected with DMEM. Lungs were collected at 2 and 4 dpi and immune responses were analyzed by RT-qPCR. (A) Immune responses at 2 dpi. (B) Immune responses at 4 dpi. N = 6 mice per group. Data are represented as mean ± s.e.m. Data was analyzed using Welch ANOVA with Dunnett's T3 multiple comparisons test, comparing each group to wild-type SARS-CoV. * $p < 0.05$, ** $p < 0.01$.

antiviral state. PLpro can remove ubiquitin and ISG15 from cellular substrates, thereby potentially manipulating a diverse array of cellular processes, including the immune response. To gain insight into the contribution of the DUB and deISGylating activities of PLpro to immune evasion during infection, these functions need to be separated from its crucial role in replicase polyprotein cleavage. This can be achieved by introducing substitutions into the ubiquitin- and ISG15-binding domains of PLpro, which are distant from the catalytic core of the protease, thereby preventing ubiquitin or ISG15 binding while leaving the catalytic site intact. Several biochemical studies have proposed specific mutations that can disrupt either the DUB or the deISGylating activity (or both) for coronavirus PLpro proteins [15,16,29,31,35,37]. Based on these studies and the crystal structures of PLpro in complex with K48-linked diubiquitin or ISG15, we introduced mutations to disrupt ubiquitin or ISG15 binding. By introducing these mutations into the SARS-CoV-2 genome, we were able, for the first time, to study the role of the DUB activity in the context of the complete viral replication cycle. Our findings show that DUB mutant viruses display similar growth kinetics as wild-type SARS-CoV-2, both in human lung cells and in mice (Figs 3 and 5). Furthermore, disruption of the DUB activity did not affect the lethality of the infection in mice (Fig 5). Interestingly, our DUB mutant SARS-CoV-2 elicited a slightly stronger immune response in human lung cells, but not in mice (Figs 4 and 6). To our knowledge, this is the first study to directly investigate the importance of the DUB activity of PLpro during SARS-CoV-2 infection, while our laboratory previously evaluated the impact of similar mutations in MERS-CoV [36]. Together, these PLpro studies highlight major differences in the contribution of the DUB activity of PLpro to the replication and immune evasion of highly pathogenic coronaviruses.

Several biochemical studies have shown that the SARS-CoV-2 PLpro domain hydrolyses ISG15 conjugates more efficiently than K48-linked ubiquitin, while SARS-CoV PLpro preferentially cleaves K48-linked ubiquitin [14–16,37,45]. A recent study comparing the PLpro proteins of all human coronaviruses suggested that a preference for ISG15 over ubiquitin is associated with a stronger suppression of the immune response [46]. Interestingly, we found that disrupting the SARS-CoV PLpro DUB activity modestly reduced viral replication and immune response *in vivo* (Figs 7 and 8), which was in contrast with SARS-CoV-2 DUB mutants displaying similar replication kinetics and immune responses as the wild-type virus (Figs 5 and 6). This seems to be in line with the *in vitro* data showing that SARS-CoV PLpro is more active as a DUB than as a deISGylase, while SARS-CoV-2 PLpro is more active as a deISGylase, which we could not disrupt without losing virus viability.

For MERS-CoV, the impact of disrupting PLpro DUB and deISGylating activities on virus replication was shown to be much stronger [36]. Introduction of a mutation (V1691R) that abrogated both the DUB and the deISGylating activity of MERS-CoV PLpro [35] led to strong attenuation of the virus in mice, which was characterized by the absence of clinical symptoms, even in the presence of virus replication in the lung [36]. Moreover, this DUB/deISGylase mutant MERS-CoV elicited a more effective innate and adaptive immune response, which protected against a challenge with a lethal dose of wild-type MERS-CoV [36]. Although we do not know how a deISGylase-competent but DUB-negative mutant MERS-CoV would have behaved, the enormous impact of this mutation in the MERS-CoV context hints at important differences between the relevance of PLpro DUB activity in SARS-CoV, SARS-CoV-2, and MERS-CoV infection. This is supported by the notion that the DUB activity of MERS-CoV PLpro is much more promiscuous than those of SARS-CoV and SARS-CoV-2 PLpro, at least judging from *in vitro* experiments. While SARS-CoV and SARS-CoV-2 PLpro can only process K48-linked ubiquitin chains efficiently, MERS-CoV PLpro cleaves all linkage types, except for M1, and can also remove mono-ubiquitin [15,30,47,48]. This implies that MERS-CoV PLpro can target a much broader substrate range, potentially explaining why its DUB activity

is so much more important during infection. Moreover, the immune response is mainly regulated by signaling involving linear and K63-linked chains [49,50], which cannot be removed by SARS-CoV and SARS-CoV-2 PLpro. This could explain why disrupting the DUB activity of these PLpro proteins only has a minor impact on the immune response in human lung cells (Fig 4), which contrasts with the strongly enhanced immune response elicited by DUB mutant MERS-CoV [36]. Since K48-linked ubiquitin chains have an important role in regulating protein degradation, it is likely that the role of the DUB activity of SARS-CoV and SARS-CoV-2 PLpro is mainly to manipulate proteostasis. For example, PLpro might also protect viral proteins from proteasome-mediated degradation, as was shown recently for a papain-like protease expressed by the distantly related arterivirus porcine reproductive and respiratory syndrome virus [51]. The DUB mutant viruses generated in this study could serve as a tool to study the effect on the ubiquitination of viral proteins and potential deubiquitination by PLpro. Furthermore, these DUB mutant viruses can give us insights into the cellular antiviral mechanisms that are regulated by ubiquitination. Several studies have investigated the ubiquitinated proteome of SARS-CoV-2-infected cells (Stukalov *et al*, 2021; Zhang *et al*, 2021; Xu *et al*, 2022). However, since ubiquitination is counteracted by PLpro, it could be that important substrates are not identified when using wild-type SARS-CoV-2. It would thus be highly interesting to compare the ubiquitinated proteome of cells infected with wild-type or DUB mutant SARS-CoV-2 to identify crucial targets for ubiquitination during infection.

The attenuated phenotype of DUB mutant MERS-CoV could provide a basis for the development of a live-attenuated vaccine. Therefore, we aimed to translate these findings to SARS-CoV-2. Even though efficacious vaccines were developed at record speed, a live-attenuated vaccine would still be very beneficial as it is likely to provide more robust and longer-lasting immunity, characterized by broad T-cell and humoral responses to a wide range of viral proteins [52]. However, disrupting the DUB activity of SARS-CoV-2 PLpro alone is clearly not enough to attenuate the virus. It is highly likely that the deISGylating activity of SARS-CoV-2 PLpro needs to be disrupted to attenuate the virus. This may be technically challenging, as our attempts did not yield any viable virus mutants when targeting this activity.

ISG15 is an important regulator of the innate immune response and ISGylation is manipulated by many viruses, including SARS-CoV-2 [28]. By its deISGylating activity SARS-CoV-2 PLpro removes ISG15 from MDA5, thus inhibiting activation of the IFNβ response [27]. We could only identify two combinations of mutations, W106L/A107S and R166S/E167R, that abrogated deISGylating activity, but unfortunately introduction of these mutations into the virus did not yield viable progeny. These residues are located close to the active site and processing of the viral replicase polyprotein was strongly reduced in an *in vitro* assay. Therefore, alternative mutations need to be identified that reduce deISGylating activity and can be incorporated into the virus. The mutations that we introduced into the PLpro S2 site, which binds to the N-terminal domain of ISG15, had hardly any effect on deISGylating activity (Fig 2C and 2D). Only one mutant exhibited a 33% reduction in deISGylating activity. A potential explanation for this is provided by a study which showed that the N-terminal domain of ISG15 is only weakly bound by PLpro, which was also accompanied by conformational flexibility [53]. Thus, one likely needs to disrupt the binding of the PLpro S1 site to the C-terminal domain of ISG15 in order to remove its deISGylating activity. Based on our *in vitro* data, it would be interesting to introduce M208R into the virus, since this mutations reduced the deISGylating activity of PLpro by 50% (Fig 2C and 2D). However, it needs to be investigated whether this reduction will have an effect *in vivo* or whether there is enough residual activity for PLpro to counteract ISGylation. Biochemical studies identified N156, S170, Y171, and Q174 as interactors of ISG15 [15,37]. Further investigations are needed to determine whether incorporation of these mutations may yield a viable virus mutant that lacks deISGylating activity.

Although the DUB-reducing SARS-CoV-2 PLpro mutations did not lead to attenuation of the virus on their own, it could still be useful to incorporate them—in combination with other mutations—in an attenuated modified live virus vaccine. This is exemplified by a temperature-sensitive mutant of murine hepatitis virus (MHV). Mutations in the MHV macrodomain, which also resides in nsp3, attenuated the virus at nonpermissive temperatures, but these mutations had a high reversion frequency [54]. When mutations were incorporated in the papain-like protease 2, these stabilized the macrodomain mutations, although independently they did not attenuate the virus. This underlines the importance of the interplay between different nsp3 domains and suggests that PLpro mutations can contribute to an attenuated phenotype when combined with mutations in other domains. Further research is needed to elucidate whether this also holds true for attenuating mutations in other viral proteins. To date, many SARS-CoV-2 proteins have been shown to subvert the immune response in some way [7]. Therefore, in the case of PLpro mutants with reduced DUB activity, other viral proteins might be able to compensate for the reduced suppression of the immune response. It would thus be highly interesting to investigate whether combinations of mutations in different viral proteins could yield an attenuated virus that could serve as a basis for a live-attenuated vaccine.

Because of its crucial role in viral polyprotein processing, the coronavirus PLpro is an important target for the development of antiviral drugs (reviewed in [55]) and live-attenuated vaccines, as illustrated by recent findings of our laboratory on MERS-CoV [36]. Understanding the molecular mechanisms by which PLpro manipulates the host response can contribute to the rational design of antivirals, as illustrated by the development of ubiquitin variants that potently inhibit MERS-CoV and SARS-CoV-2 replication in cell culture [56,57]. Furthermore, the immune-evasive properties of PLpro could contribute to the molecular mechanisms underlying the dysregulated immune response as observed in critically ill patients [6,58]. In summary, our study provides evidence that the DUB activity of SARS-CoV-2 PLpro does not influence virus replication or innate immune responses *in vivo*. Furthermore, we showed that DUB mutant SARS-CoV is mildly attenuated, as illustrated by a reduction in virus replication and a diminished immune response during the initial phase of the infection, while our laboratory has previously shown that DUB/deISGylase mutant MERS-CoV is strongly attenuated. This highlights important differences in the mechanisms by which the PLpro domains of various coronaviruses manipulate host defense mechanisms, which should be taken into consideration during the development of therapeutics.

## Materials and methods

### Ethics statement

All animal experiments have received approval from the Animal Welfare Body of the LUMC (Leiden University Medical Center) and were performed according to the Dutch Experiments on Animals Act and the recommendations and guidelines set by the Animal Welfare Body of the LUMC, and were in strict accordance with EU regulations (2010/63/EU).

### Cells, viruses, and plasmids

HEK293T cells (ATCC, #CRL-3126) were cultured in Dulbecco's modified Eagle's medium (DMEM, Gibco) supplemented with 10% fetal calf serum (FCS, Bodinco BV), 2 mM L-glutamine, 50 IU/ml penicillin, and 50 µg/ml streptomycin (Sigma-Aldrich). Calu-3 cells (ATCC, #HTB-55) were cultured in minimum essential medium with Earle's salts (EMEM, Gibco) supplemented with 10% FCS, 2 mM L-glutamine, 0.1 mM non-essential amino acids (Lonza), 1 mM sodium pyruvate (Thermo Fisher Scientific), 50 IU/ml penicillin, and 50 µg/ml

streptomycin. VeroE6 cells were cultured in DMEM supplemented with 8% FCS, 50 IU/ml penicillin, and 50 μg/ml streptomycin. BHK-21 cells were cultured in Glasgow's MEM (Gibco) supplemented with 8% FCS, 10% tryptose phosphate broth (Thermo Fisher Scientific), 10 mM HEPES pH 7.4 (Lonza), 50 IU/ml penicillin, and 50 μg/ml streptomycin. All cell lines were cultured at 37˚C in 5% $CO_2$.

Virus infections were done in EMEM supplemented with 2% FCS, 50 IU/ml penicillin, 50 μg/ml streptomycin, and cell line specific supplements similar to the culture medium.

SARS-CoV-2/human/NLD/Leiden-0008/2020 (SARS-CoV-2/Leiden-0008) was isolated from a nasopharyngeal swab of a positively tested individual in March 2020. The complete genome sequence of this isolate is available under GenBank accession number MT705206.1. Compared to the original Wuhan isolate, this isolate contains the D614G mutation in the spike protein and three additional non-silent mutations, namely C12846U in nsp9 (A54V), C14408U in nsp12 (P323L), and C18928U in nsp14 (P267S). All experiments involving live SARS-CoV or SARS-CoV-2 were performed in a biosafety level 3 (BSL-3) laboratory.

The following plasmids were described elsewhere: pcDNA3.1(-) (Invitrogen), pcDNA-eGFP [59], pCAGGS-V5-hISG15-GG [60], pCAGGS-HA-hUbE1L [61], pCMV2-FLAG-UbcH8 [61], CS111-hHerc5-HA [60], pLuc-IFN-β [62], pRL-TK (Promega), and pEBG-RI-G-I$_{(2CARD)}$ [63]. pRK5-HA-Ubiquitin was a kind gift from Ted Dawson (Addgene, #17608) [64].

## Construction of SARS-CoV PLpro and SARS-CoV-2 PLpro mammalian expression plasmids

To generate plasmids for expression of SARS-CoV and SARS-CoV-2 PLpro in mammalian cells, synthetic DNA fragments encoding the mammalian codon-optimized PLpro sequences of SARS-CoV (amino acids 1541–1855 of pp1a) and SARS-CoV-2 (amino acids 1564–1878 of pp1a) with a V5 epitope tag at the C-terminus were ligated into pCR2.1-TOPO (Thermo Fisher Scientific). Subsequently, PLpro-V5 was cloned into the pcDNA3.1(-) vector using restriction cloning. The resulting pcDNA3.1-PLpro (SARS-CoV)-V5 and pcDNA3.1-PLpro (SARS-CoV-2)-V5 plasmids were then used as a template for the introduction of mutations using site-directed mutagenesis. For consistency with the published literature, engineered mutations are numbered according to their position in the PLpro domain.

## Generation of PLpro mutant viruses

SARS-CoV and SARS-CoV-2 PLpro mutations were introduced into the respective viruses by reverse genetics. In-yeast assembly of a SARS-CoV-2 full-length cDNA clone by transformation-associated recombination (TAR) was described previously [65]. Based on this strategy, we generated a cDNA clone containing a full-length cDNA copy of the genome of the SARS-CoV-2/Leiden-0008 clinical isolate. Overlapping DNA fragments that span the entire SARS-CoV-2 genome were amplified in plasmids and used as templates to generate PCR fragments that were then assembled into the pCC1BAC-his3 vector by TAR in *S. cerevisae*. The sequence of the resulting full-length cDNA clone (pCC1BAC-his3-SARS-CoV-2/L-0008) was verified by next-generation sequencing. The plasmid containing the PLpro sequence was used as a template to introduce the designed substitutions by site-directed mutagenesis. Subsequently, PCR products were generated from the mutated fragment and this fragment was then incorporated to generate PLpro mutant SARS-CoV-2 cDNA clones. These were isolated from yeast and subjected to multiplex PCR targeting all the assembly junctions, as described previously [65]. Clones that were positive for all junctions were transformed into *E. coli* for large-scale

preparations of the bacmids. The sequence of the cDNA clones and presence of the introduced substitutions were verified using next-generation sequencing.

SARS-CoV mutant viruses were generated by two-step en passant recombineering. Substitutions in the PLpro-coding region were introduced into a BAC containing a full-length cDNA copy of the genome of the SARS-CoV Frankfurt-1 clinical isolate, as described previously [66].

pBAC-SARS-CoV and pCC1BAC-his3-SARS-CoV-2/L-0008 wild-type en PLpro mutant cDNA clones were isolated from bacteria and linearized using NotI. Linearized DNA was used as a template for *in vitro* transcription using the mMESSAGE mMACHINE T7 transcription kit (Thermo Fisher Scientific). *In vitro*-transcribed RNA was electroporated into $5 \times 10^6$ BHK-21 cells using the Amaxa Nucleofector 2b, program A-031, with the Cell Line Nucleofector Kit T (Lonza). BHK-21 cells were used because they can easily be electroporated with the viral RNA and this yields a first generation of virus particles. However, these cells cannot further support receptor mediated entry of produced virus particles. Therefore, electroporated BHK-21 cells were immediately resuspended in prewarmed medium and mixed with $5 \times 10^6$ VeroE6 cells. Cells were incubated at 37°C and virus-containing supernatant was collected when extensive cytopathic effect was observed. Viruses were passaged twice on Calu-3 cells to grow stocks for further experiments, while maintaining the integrity of the furin cleavage site in the Spike protein [67]. After every passage, the presence of the engineered substitutions was confirmed by Sanger sequencing. Virus titers were determined on VeroE6 cells by plaque assay as described previously [57].

## Bacterial expression and purification of PLpro

To generate a plasmid for bacterial expression of SARS-CoV-2 PLpro, the sequence encoding bacterial codon-optimized PLpro (amino acids 1564–1878 of pp1a) was cloned into the bacterial expression vector pET16b in frame with a C-terminal His$_6$-tag. The resulting pET16b-PLpro-His$_6$ plasmid was then used as a template for the introduction of mutations using site-directed mutagenesis.

To express SARS-CoV-2 PLpro, *E. coli* Rosetta2(DE3) cells were transformed with pET16b-PLpro-His$_6$. Cells were grown at 37°C to an OD$_{600}$ of 0.7, upon which protein expression was induced overnight at 18°C using 0.4 mM isopropyl β-D-1-thiogalactopyranoside (IPTG). Cell pellets were resuspended in lysis buffer (25 mM Tris pH 8.0, 200 mM NaCl, 5 mM imidazole, 1 mM TCEP) and lysed by sonication. PLpro was then affinity purified using nickel beads. Proteins were eluted in 25 mM Tris pH 8.0, 500 mM NaCl, 250 mM imidazole, 1 mM TCEP. Eluates were concentrated and injected onto a S200 gel filtration column for size-exclusion chromatography. Peak fractions corresponding to PLpro were pooled and concentrated and purified PLpro was stored in 25 mM Tris pH 8.0, 200 mM NaCl, 1 mM TCEP.

## Biochemical characterization of the DUB, deISGylating, and polyprotein processing activity of PLpro

The assays involving AMC substrates were performed in non-binding-surface, flat bottom, low-flange, black 384-well plates (Corning 3820) at ambient temperature in a buffer containing 20 mM Tris pH 7.5, 150 mM NaCl, and 5 mM DTT. The DUB activity of the SARS-CoV-2 PLpro mutants was characterized by assessing the hydrolysis of Ub-AMC and K48-diUb-AMC (synthesized in-house [68,69]). Ub-AMC was used at final concentrations of 50, 30, 20, 10, 5, 2.5, and 1.25 μM. For K48-diUb-AMC, a 2-fold serial dilution was made starting at 40 μM. The reaction was initiated by the addition of PLpro at a final concentration of 50 nM (for Ub-AMC) or 20 nM PLpro (for K48-diUb-AMC). AMC release was monitored for 1 h

taking a measurement every minute and normalized to a standard curve that was generated by incubating known concentrations of Ub-AMC with 100 nM of the DUB ubiquitin C-terminal hydrolase L3 (UCHL3) or K48-diUb-AMC with 100 nM PLpro until the substrates were completely converted. The linear portion of the curve was used to calculate initial velocities, which were then plotted against the substrate concentration. Kinetic parameters were calculated using the $k_{cat}$ function in GraphPad Prism with enzyme concentration constraint to 50 nM (for Ub-AMC) or 20 nM (for K48-diUb-AMC).

To assess the deISGylating activity of the PLpro mutants, a 2-fold serial dilution of PLpro was made starting at 40 nM. The reaction was initiated by addition of ISG15-AMC (R&D systems) at a final concentration of 150 nM. AMC release was normalized to a standard curve that was generated by incubating known concentrations of ISG15-AMC with murine ubiquitin-specific peptidase 18 (mUSP18) [70]. The linear portion of the curve was used to calculate initial velocities, which were then plotted against the PLpro concentration. The slope of the curve was used to estimate the apparent rate constant of ISG15-AMC hydrolysis.

All measurements were performed in triplicate using a PHERAstar microplate reader (BMG Labtech).

The polyprotein processing activity of PLpro was assessed in a FRET assay using a FRET peptide with the following sequence: Dabcyl-KTLKGGAPTKVTE-EDANS. TLKGGAPTKVT corresponds to the nsp2-3 junction of the SARS-CoV-2 polyprotein. The FRET peptide was synthesized at the Peptide and MHC-tetramer facility of the LUMC. PLpro activity was assessed by incubating 2 μM PLpro with 50 μM FRET peptide in 20 mM Tris pH 7.5, 200 mM NaCl, 0.1 mM EDTA, 1 mM TCEP, 12.5% DMSO. The assay was performed in flat bottom, black 96-well plates (Greiner Bio-One 655209) at 37°C for 1 h with measurements taken every minute. Measurements were acquired using an EnVision multilabel plate reader (Perkin Elmer). Initial velocities were calculated from the linear portion of the curve as a measure for the substrate cleavage rate.

## Cell-based DUB and deISGylation assays

For the DUB assay, 60–80% confluent HEK293T cells were transfected with plasmids encoding HA-ubiquitin (500 ng), eGFP (250 ng), and SARS-CoV-2 PLpro-V5 (500 ng). Empty vector was added to equalize the total amount of DNA to 1.6 μg per well (12-well format). DNA mixes were prepared in Opti-MEM (Gibco) and mixed with linear polyethylenimine (PEI 25K, Polysciences) that was prediluted in Opti-MEM in a 1:3 DNA-to-PEI ratio. Transfection mixes were incubated for 20 minutes at room temperature and then added to the cells in a dropwise manner. At 24 hours post transfection, transfection efficiency was assessed based on the GFP signal and cell lysates were harvested in 2x Laemmli Sample buffer (2x LSB, 250 mM Tris pH 6.8, 4% SDS, 20% glycerol, 10 mM DTT, 0.01% bromophenol blue) and heated for 10 minutes at 95°C.

For the deISGylation assay, 60–80% confluent HEK293T cells were transfected with plasmids encoding V5-hISG15 (250 ng), HA-hUbe1L (250 ng), FLAG-UbcH8 (250 ng), hHerc5-HA (250 ng), eGFP (250 ng), and SARS-CoV-2 PLpro-V5 (35 ng). Transfection mixes were prepared in a similar manner as for the DUB assay. Transfected cells were incubated at 37°C for 48 hours and medium was refreshed after 24 hours. Cell lysates were harvested in 2x LSB and heated for 10 minutes at 95°C.

## IFNβ luciferase reporter assay

IFNβ promoter activity was assessed using a luciferase reporter assay. 60–80% confluent HEK293T cells were transfected with plasmids encoding pLuc-IFNβ (50 ng), pRL-TK (5 ng),

RIG-I$_{(2CARD)}$ (50 ng), and SARS-CoV or SARS-CoV-2 PLpro-V5 (695 ng). Where appropriate, empty vector was added to bring the total DNA amount per well (24-well format) to 800 ng. DNA was mixed with PEI in a 1:3 DNA-to-PEI ratio and incubated for 20 minutes at room temperature, after which transfection mixes were added to the cells in a dropwise manner. At 24 hours post transfection, luciferase activity was measured using the Dual-Luciferase Reporter Assay System (Promega). Firefly luciferase values were normalized to *Renilla* luciferase values to correct for variations in the transfection efficiency. All data is shown relative to the cells that were transfected with RIG-I$_{2CARD}$ only. Cell lysates were mixed with 4x LSB (500 mM Tris pH 6.8, 8% SDS, 40% glycerol, 20 mM DTT, 0.02% bromophenol blue) in a 3:1 ratio, so that expression of transfected proteins could be analyzed by western blot.

## Western blot analysis

For the analysis of DUB and deISGylation assays and of ISG15 expression in infected cells, proteins were separated by SDS-PAGE and transferred onto Immobilon-FL PVDF membrane (Merck Millipore). Membranes were blocked in 1% casein in PBS with 0.05% Tween-20 (PBST) for 1 hour at room temperature, followed by overnight incubation with primary antibody at 4°C. The following primary antibodies were used: α-β-actin (clone AC74, Sigma, #A5316), α-HA (clone HA.C5, Abcam, #ab18181), α-α-tubulin (clone B-5-1-2, Sigma-Aldrich, #T5168), α-V5 (2F11F7, Thermo Fisher/Invitrogen, #37–7500), α-GFP (042150, Leiden [59]), α-ISG15 (clone F-9, Santa Cruz, sc-166755), and polyclonal rabbit anti-Membrane (M) protein of SARS-CoV (cross-reacts with SARS-CoV-2 M protein [71]). Membranes were subsequently incubated with mouse- or rabbit-specific horseradish peroxidase-coupled secondary antibodies (Agilent Dako, P0447 and P0217), after which proteins bands were detected using Clarity Western ECL substrate (BioRad) and visualized on an Alliance Q9 Advanced imaging system (Uvitec).

## SARS-CoV-2 growth kinetics and immune response

Growth kinetics of SARS-CoV-2 wild-type virus and PLpro mutants were determined by inoculating confluent monolayers of Calu-3 cells at a multiplicity of infection (MOI) of 0.1, based on the titers that were determined on VeroE6 cells. Inocula were prepared in EMEM with 2% FCS and left on the cells for 2 hours at 37°C. Cells were washed three times with PBS before adding EMEM with 2% FCS. Supernatants as well as cells lysed in TriPure Isolation reagent (Roche Diagnostics) were harvested at 4-hour intervals from 8 to 48 hpi. Cell lysates were used to determine intracellular viral RNA copy numbers and host immune responses by RT-qPCR. 100 μl supernatant was mixed with 900 μl TriPure spiked with RNA of equine arteritis virus (EAV) to determine extracellular viral RNA copy numbers by RT-qPCR. Virus titers in the supernatant were determined by plaque assays on VeroE6 cells.

## Infection of K18-hACE2 mice with SARS-CoV and SARS-CoV-2 wild-type and DUB mutants

All animal experiments were performed in K18-hACE2 transgenic mice that express the human ACE2 receptor under control of the cytokeratin 18 (K18) promoter [39]. Animals were obtained from the Jackson Laboratory (B6.Cg-Tg(K18-ACE2)2Prlmn/J) and bred in-house. Mice were housed in individually ventilated isolator cages under specific pathogen-free conditions in a temperature-controlled room with 12h light-dark cycle and *ad libitum* access to water and food. All experiments were performed at the animal BSL-3 (ABSL-3) facility of the LUMC.

To verify that recombinant SARS-CoV-2/L-0008 (rSARS-CoV-2) had similar lethality as the clinical isolate, a dosing experiment was performed in 8–16 weeks-old male and female mice. Infected groups consisted of 9 mice, while the control group had 8 mice. Mice were anesthetized with isoflurane gas and inoculated intranasally with $2.5x10^4$ or $1x10^5$ pfu of SARS-CoV-2 (isolate) or rSARS-CoV-2 in a total volume of 50 µl DMEM. Control mice were inoculated with 50 µl DMEM. Mice were monitored twice per day and bodyweight and clinical discomfort were recorded daily. Humane endpoints were defined as weight loss of ≥20% of the original bodyweight prior to inoculation and/or a moribund state. When reaching the humane endpoint, mice were euthanized by intraperitoneal injection of 0.8 mg/g bodyweight sodium pentobarbital under isoflurane anesthesia. Both doses of rSARS-CoV-2 were 100% lethal and there was no difference in loss of bodyweight (S7 Fig). Both doses of SARS-CoV-2 (isolate) had 80% lethality, and the difference with rSARS-CoV-2 was not significant.

To assess the lethality of the SARS-CoV and SARS-CoV-2 DUB mutants, survival experiments were performed. The mice for the SARS-CoV experiments were 19–22 weeks old, while the mice for the SARS-CoV-2 experiments were 11–14 weeks old. Each group contained 10–11 male and female mice per group, except for control groups, that contained 8 mice. SARS-CoV and SARS-CoV-2 experiments were performed independently. Mice were anesthetized with isoflurane and inoculated intranasally with $1x10^4$ pfu SARS-CoV wild-type, F70S, or M209S or SARS-CoV-2 wild-type, F69S, F69R, or M208S. Control mice were inoculated with 50 µl DMEM. We used a dose of $1x10^4$ pfu as previous dosing experiments showed that this was the lowest SARS-CoV-2 dose that had a consistent lethality rate of ≥80%. Mice were monitored twice per day and bodyweight and clinical discomfort were recorded daily. Upon reaching the humane endpoint, mice were euthanized by intraperitoneal injection of 0.8 mg/g bodyweight sodium pentobarbital under isoflurane anesthesia.

To assess virus replication and immune responses, similar experiments were done as the survival experiments. However, at 2 and 4 dpi, 6 mice per group were sacrificed and lungs were dissected. One lung lobe was instilled intratracheally with 4% paraformaldehyde and fixed for 24 h by immersion in 4% paraformaldehyde, after which it was transferred to 70% EtOH and stored at 4°C. The other lung was divided in two parts, one part for RNA isolation and one part for virus titer determination.

## RNA isolation from the lungs of infected mice

Lung parts for RNA isolation were weighed and transferred to gentleMACS M tubes (Miltenyi Biotec) containing 2 ml TriPure. Tissue was homogenized using a gentle MACS Dissociator, program RNA_02 (Miltenyi Biotec). Homogenates in gentleMACS M tubes were centrifuged for 1 min at 250 x g, transferred to 2 ml screwcap vials, and centrifuged for 5 minutes at 9,500 x g, after which supernatants were collected. To extract RNA, chloroform was added and liquid phases were separated by centrifugation. RNA was precipitated from the aqueous phase using isopropanol. Viral copy numbers were determined by RT-qPCR. To verify the presence of the introduced PLpro substitutions, RNA that was extracted from lungs harvested at 4 dpi was reverse transcribed into cDNA using RevertAid H minus reverse transcriptase (Thermo Fisher Scientific) and random hexamers (Promega). The PLpro region was amplified by PCR and the PCR product was analyzed by Sanger sequencing.

## Determination of virus titers in the lungs of infected mice

Lung parts for virus titer determination were weighed and transferred to gentleMACS M tubes (Miltenyi Biotec) containing 2 ml PBS with 100 units/ml penicillin, 100 units/ml streptomycin (Lonza), 50 µg/ml gentamycin (Sigma-Aldrich), and 0.25 µg/ml Fungizone (Gibco). Tissue

was homogenized using a gentle MACS Dissociator, program Lung_02 (Miltenyi Biotec). Homogenates in gentleMACS M tubes were centrifuged for 1 min at 250 x g, transferred to 2 ml screwcap vials, and centrifuged for 5 minutes at 9,500 x g, after which supernatants were collected. Virus titers were determined by plaque assays on VeroE6 cells and expressed as pfu/ g lung.

### Determination of viral RNA copy numbers by RT-qPCR

RNA was isolated from cell lysates and supernatants by liquid phase separation using TriPure and chloroform. Viral RNA copy numbers in cell lysates and supernatants and in lung homogenates were measured using the TaqMan Fast Virus 1-step master mix (Thermo Fisher Scientific) on a CFX384 Touch Real-Time PCR Detection System (BioRad) with the following program: 5 min at 50˚C, 20 s at 95˚C, followed by 45 cycles of 5 s at 95˚C and 30 s 60˚C. We used primer and probes sets against the RdRp and E genes that were modified based on previously described primer and probe sets [72]. Sequences can be found in S2 Table. Primers against RdRp only detect genomic viral RNA, while primers against E detect genomic and subgenomic viral RNA. Primers and probes against RdRp were different for SARS-CoV and SARS-CoV-2, while primers and probe for E were the same for both viruses. A standard curve of 10-fold serial dilutions of known quantities of *in vitro*-transcribed RNA containing the target sequences was included for absolute quantification of viral RNA copy numbers. Intracellular viral RNA levels were normalized to hPGK1 expression, as determined by PGK1 TaqMan gene expression Assay Vic-MGB (Thermo Fisher Scientific, #4448490, assay ID Hs00943178_g1). Extracellular viral RNA levels were normalized to EAV expression. Viral RNA levels in the lungs of infected mice were normalized to mPGK1 expression, which was determined by PGK1 TaqMan gene expression Assay Vic-MGB (Thermo Fisher Scientific, #4448490, assay ID Mm04213280_s1). Viral RNA levels in the lung were corrected for the weight of the piece of lung that was homogenized.

### Evaluation of immune responses in infected cells and mouse lungs by RT-qPCR

To determine the expression of genes related to the immune response in Calu-3 cells infected with SARS-CoV-2 or lungs of mice infected with SARS-CoV or SARS-CoV-2, 1 μg of RNA was reverse transcribed into cDNA using RevertAid H minus reverse transcriptase (Thermo Fisher Scientific) and random hexamers (Promega). Gene expression was determined by RT-qPCR using iQ SYBR Green supermix (BioRad) on a CFX384 Touch Real-Time PCR Detection System (BioRad) with the following program: 3 min at 95˚C, 30 s at 60˚C, 40 cycles of 10 s at 95˚C, 10 s at 60˚C, and 30 s at 72˚C, followed by 10 s at 95˚C and melt curve analysis using a temperature gradient from 60˚C to 95˚C with 0.5˚C increment. A list of all the genes measured and the corresponding primers can be found in S3 Table. mRNA expression was quantified as arbitrary units that were deduced from a standard curve of 5-fold serial dilutions that was included in the RT-qPCR. mRNA expression in Calu-3 cells was normalized to RPL13A and ACTINB and fold change over SARS-CoV-2 wild-type was calculated. mRNA expression in the lungs of infected mice was normalized to mPGK1 expression and expressed as fold change over mock.

### Statistical analysis

Statistical analyses were performed in GraphPad Prism (version 9). All data are represented as mean ± s.e.m., unless stated otherwise. Survival experiments were analyzed using log-rank (Mantel-Cox) test. One-way ANOVA and Dunnett's multiple comparison test were used for

all other comparisons. In the multiple comparison test, each group was compared to the wild-type virus for infection experiments and to the wild-type PLpro for luciferase reporter assays. Gene expression data on immune responses in infected Calu-3 cells were analyzed by mixed-effects analysis with Dunnett's multiple comparisons test. Gene expression data on immune responses in the lungs of infected mice were analyzed by Welch ANOVA with Dunnett's T3 multiple comparisons test, since the assumptions for one-way ANOVA were not met. Differences were considered statistically significant when $p<0.05$. All p-values were two-sided.

## Supporting information

**S1 Fig. Crystal structure of SARS-CoV-2 PLpro in complex with mouse ISG15.** (A) Cartoon representation of SARS-CoV PLpro (aquamarine) in complex with K48-linked diubiquitin (grey; PDB 5E6J) aligned with SARS-CoV-2 PLpro (sky blue) in complex with mouse ISG15 (pale green; PDB 6YVA). Catalytic triad residues (C111, H272, D286) are shown in ball-and-stick representation. (B-C) Interaction of PLpro residues V66 and F69 with mISG15 residues A2, T20, and M23 (B), which is predicted to be disrupted by introduction of V66A and F69R (C). (D) Alignment of human (UniProt P05161) and mouse (UniProt Q64339) ISG15. Mouse ISG15 residues A2 and T20 are not conserved.
(TIF)

**S2 Fig. DUB and deISGylating activity of SARS-CoV-2 PLpro mutants in cell-based assays.** (A) Immunoblot analysis of ubiquitin conjugation in HEK293T cells transfected with ubiquitin and SARS-CoV-2 PLpro wild-type or mutants at 24 hours post transfection. (B) Immunoblot analysis of ISG15 conjugation in HEK293T cells transfected with ISG15 and SARS-CoV-2 PLpro wild-type or mutants at 48 hours post transfection. FEH(M) = F69S/E70K/H73G (/M208S). Representative blots of n = 5 experiments.
(TIF)

**S3 Fig. SARS-CoV-2 DUB mutant viruses induce higher ISG15 protein expression.** Calu-3 cells were infected at MOI 0.1 with wild-type or DUB mutant SARS-CoV-2 and protein lysates were collected at 24 hpi to analyze ISG15 protein expression by western blot. ISG15 expression was normalized to β-actin expression and fold changes were calculated relative to wild-type SARS-CoV-2. Representative blots of n = 2 experiments. FEH(M) = F69S/E70K/H73G (/M208S).
(TIF)

**S4 Fig. Genetic stability of SARS-CoV-2 PLpro mutations *in vivo*.** (A-B) Genetic stability of F69S and F69R (A) and M208S (B) *in vivo* was determined by RT-PCR to amplify the PLpro-coding region from RNA extracted from lung homogenates harvested at 4 dpi from infected mice. The PCR product was Sanger sequenced. 2–3 mice were analyzed per virus.
(TIF)

**S5 Fig. Disruption of the DUB activity of SARS-CoV but not SARS-CoV-2 PLpro induces a stronger IFNβ response.** (A) Relative DUB and deISGylating activity of SARS-CoV and SARS-CoV-2 PLpro mutants compared to wild-type PLpro as assessed by the hydrolysis of Ub-AMC, K48-diUb-AMC, and ISG15-AMC. (B) IFNβ promoter-driven luciferase activity in HEK293T cells transfected with *Renilla* and firefly luciferase, wild-type or mutant SARS-CoV or SARS-CoV-2 PLpro, and constitutively active RIG-I$_{2CARD}$. Luciferase activity was determined at 24 hours post transfection. Firefly luciferase was normalized to *Renilla* luciferase and expressed relative to the positive control, RIG-I$_{2CARD}$. Panels below show representative immunoblots using V5 to verify expression of PLpro and α-tubulin as loading control. [a] Data

in panel A for SARS-CoV PLpro is from Békés et al. [31]. Data for SARS-CoV-2 PLpro was taken from Fig 2. Data in B are represented as mean ± s.e.m. of 3 experiments. One-way ANOVA with Dunnett's multiple comparisons test, comparing each group to wild-type PLpro. * $p < 0.05$, ** $p < 0.01$, *** $p < 0.001$, **** $p < 0.0001$.
(TIF)

**S6 Fig. Genetic stability of SARS-CoV PLpro mutations *in vivo*.** (A-B) Genetic stability of F70S (A) and M209S (B) *in vivo* was determined by RT-PCR to amplify the PLpro-coding region from RNA extracted from lung homogenates harvested at 4 dpi from infected mice. The PCR product was Sanger sequenced. 2–3 mice were analyzed per virus. Silent mutations in A69 and Y208 are marker mutations that were introduced on purpose to exclude contamination with the parental virus.
(TIF)

**S7 Fig. Lethality of recombinant and clinical isolate of SARS-CoV-2 in K18-hACE2 mice.** K18-hACE2 mice were infected intranasally with $2.5 \times 10^4$ or $1 \times 10^5$ pfu rSARS-CoV-2 or SARS-CoV-2 (isolate) or mock-infected with DMEM. (A) Survival curves (%). (B) Bodyweight loss (% from initial weight). Dashed line indicates 20% weight loss upon which mice are euthanized. n = 8 mice per group for mock and n = 9 for SARS-CoV-2-infected groups. Log-rank test for survival analysis (A). One-way ANOVA with Šídák's multiple comparisons test (B). ns: not significant, * $p < 0.05$, ** $p < 0.01$, *** $p < 0.001$, **** $p < 0.0001$.
(TIF)

**S1 Table. Kinetic parameters of SARS-CoV-2 PLpro mutants on AMC substrates.**
(XLSX)

**S2 Table. RT-qPCR primers and probes for viral genes.** [a] Sarbeco primers and probes detect SARS-CoV and SARS-CoV-2 viral RNA, while SARS-CoV-2 primers and probes have enhanced specificity towards SARS-CoV-2 viral RNA. [b] W is A/T; R is G/A; M is A/C; S is G/C. FAM: 6-carboxyfluorescein; BHQ1/2: black hole quencher 1/2.
(XLSX)

**S3 Table. RT-qPCR primers for cellular genes.**
(XLSX)

**S1 Data. Raw data.**
(XLSX)

## Acknowledgments

We thank Peter Bredenbeek, Linda Boomaars-van der Zanden, Marissa Linger, Tim Dalebout, and Jessika Zevenhoven-Dobbe for technical support. We are grateful to Volker Thiel (Institute of Virology and Immunology, University of Bern) for providing us with the SARS-CoV-2 TAR system. We thank the Experimental Animal Facility of the LUMC for their support. We also thank Jan Wouter Drijfhout and Robert Cordfunke (Peptide and MHC-tetramer facility, LUMC, Leiden) for the synthesis of the nsp2-3 FRET peptide and Patrick Celie and Justina Kazokaite (Protein production facility, Dutch Cancer Institute, Amsterdam) for the production of recombinant SARS-CoV-2 PLpro proteins.

## Author Contributions

**Conceptualization:** Mariska van Huizen, Sebenzile K. Myeni, Marjolein Kikkert.

**Formal analysis:** Mariska van Huizen.

**Funding acquisition:** Mariska van Huizen, Marjolein Kikkert.

**Investigation:** Mariska van Huizen, Jonna R. Bloeme - ter Horst, Heidi L. M. de Gruyter, Robert C. M. Knaap, Anouk A. Leijs, Brian L. Mark, Sebenzile K. Myeni.

**Methodology:** Mariska van Huizen, Paul P. Geurink, Sebenzile K. Myeni.

**Project administration:** Mariska van Huizen, Sebenzile K. Myeni.

**Resources:** Paul P. Geurink, Gerbrand J. van der Heden van Noort, Tessa Nelemans, Natacha S. Ogando, Nadya Urakova.

**Supervision:** Eric J. Snijder, Sebenzile K. Myeni, Marjolein Kikkert.

**Validation:** Mariska van Huizen.

**Visualization:** Mariska van Huizen.

**Writing – original draft:** Mariska van Huizen.

**Writing – review & editing:** Mariska van Huizen, Paul P. Geurink, Gerbrand J. van der Heden van Noort, Robert C. M. Knaap, Tessa Nelemans, Natacha S. Ogando, Nadya Urakova, Brian L. Mark, Eric J. Snijder, Sebenzile K. Myeni, Marjolein Kikkert.

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
