## [Decision Letter · Decision Letter 0]

21 Feb 2024

Dear Ms van Huizen,

Thank you very much for submitting your manuscript "Deubiquitinating activity of SARS-CoV-2 papain-like protease does not influence virus replication or innate immune responses *in vivo*" for consideration at PLOS Pathogens. As with all papers reviewed by the journal, your manuscript was reviewed by members of the editorial board and by several independent reviewers. The reviewers appreciated the attention to an important topic. Based on the reviews, we are likely to accept this manuscript for publication, providing that you modify the manuscript according to the review recommendations.

Thank you for the submission of this manuscript. It has been reviewed by 2 reviewers and their comments are below. Overall they are very positive reviews with minor comments on additions that they believe will enhance the article. Please read through and let me know if there are any questions before resubmission. Thanks.

Sincerely,

Matthew B. Frieman

Guest Editor

PLOS Pathogens

Sonja Best

Section Editor

PLOS Pathogens

Michael Malim

Editor-in-Chief

PLOS Pathogens

orcid.org/0000-0002-7699-2064

Thank you for the submission of this manuscript. It has been reviewed by 2 reviewers and their comments are below. Overall they are very positive reviews with minor comments on additions that they believe will enhance the article. Please read through and let me know if there are any questions before resubmission. Thanks.

Reviewer Comments (if any, and for reference):

Reviewer's Responses to Questions

**Part I - Summary**

Reviewer #1: In this study the authors aimed to determine the contribution of the DUB and deISGylating activities of the SARS-CoV-2 papain-like protease to viral replication and modulation of the innate immune responses. Using a structure-guided approach, they introduced several mutations in PLpro that specifically disrupt binding to ubiquitin or ISG15 without affecting the protease activity that processes the viral polyprotein. Biochemical assays confirmed reduced DUB and/or deISGylase activity for several viable mutants. By generating recombinant SARS-CoV-2 and SARS-CoV carrying representative PLpro DUB mutations, the authors provide evidence that the DUB function is dispensable for SARS-CoV-2 replication and disease in mice, while it plays a subtle role for SARS-CoV.

General comments:

This is well-conducted study using using both in vitro and in vivo approaches to dissect the multifunctional properties of PLpro during infection. The combined use of virological, immunological and animal experiments significantly extends our understanding about the differential roles of coronavirus PLpro domains as well as key differences between highly pathogenic family members.

Reviewer #2: The manuscript by van Huizen and colleagues describes the analysis of the SARS-CoV-2 DUB and deISGylating activities residing in PLpro. Since PLpro is essential for CoV polyproteinprocessing the aim was to un-couple DUB, deISGylation from the essential PLpro activity and to study the impact of DUB and deISGylation in the context of virus infection. Structure-guided aa changes in PLpro showed for some substitutions a lack or a reduced DUB or deISGylation activities, while for some changes PLpro poly processing activity remained intact. However, the impact of these changes on viral replication in vitro and in vivo remains elusive since no major impact could be identified. This is in stark contrast to a similar approach that was used for MERS-CoV where clear impact on virus replication, particularly in vivo has been observed.

Overall, this is very good and comprehensive study that will stimulate the field. The authors appropriately discussed possible reasons why DUB or deISGylation has only minor impact on SARS-CoV-2 replication compared to MERS-CoV.

**Part II – Major Issues: Key Experiments Required for Acceptance**

Reviewer #1: Specific comments:

In Figure 3B, its strange that infectious virus particles can be detected at 8 hours post infection. The SARS-CoV-2 life-cycle takes atleast 12 hours for a single round, which would indicate that what is being detected here might be the inoculum. Along the same lines, in Figure 3A, the no of plaques appear to be significantly lower with the mutant viruses compared to the wild-type virus, yet the pfu/ml values seem to be similar in Figure 3B. Are the dilutions shown in Figure 3A different in the mutants v/s the wild-type?

In Figure 4, it would be useful to see the immune responses measured at protein levels rather than measured solely at RNA levels. The authors should therefore measure cytokines and chemokines in cell culture supernatants and mouse lung homogenates by ELISA or multiplex immunoassay. Similarly, activation of cytokine signalling pathways such as STAT1/STAT3 phosphorylation might provide better indication of whether such signalling cascades are affected.

For the mice infection studies (Figure 6 and 7), while the PLpro mutant viruses display modest defects in replication, combination with other attenuating mutations may reveal synergistic processes, which in turn might show differences in cytokine profiles. Similarly, larger group sizes may be needed to detect significance in replication differences.

Besides cytokine responses, it would be useful if the authors could also evaluate and include antibody titres in mouse serum by ELISA at various time points post-infection and CD4+ and CD8+ T cells in lungs and spleen. This would provide greater insights into the role of PLpro’s DUB activity in functional properties and kinetics of adaptive immune responses during SARS-CoV-2 infection and disease pathogenesis.

Reviewer #2: no major concerns

**Part III – Minor Issues: Editorial and Data Presentation Modifications**

Reviewer #1: Some of the figure legends do not contain adequate experimental details.

Reviewer #2: 1. Results section, biochemical characterisation (lines 179ff): Please add a few sentences to introduce the experimental setup. It is unclear for the reader at first glance (without digging into the methods section) how PLpro was expressed and used for the assays.

2. Figure 1A: A simple illustration of PLpro in relation to the viral genome and depiction of introduced changes would be good to have in addition to the structure.

3. In vivo experiments. While the authors did not see major differences in viral load (RNA and titers) and only minor differences in some cases for host responses, it might be interesting to include pathology. In some cases (Fig 5B) recovery from weight loss is delayed compared to wt. Together with the observation that some cytokines are slightly increased for some mutants, it might be good to check if this may impact disease/lung pathology. If those data are available it would be good to add them.

4. Is it possible to check for DUB or deISGylation activity of mutant viruses versus wt virus in infected cells?

PLOS authors have the option to publish the peer review history of their article (what does this mean?). If published, this will include your full peer review and any attached files.

Reviewer #1: **Yes: **Sumana Sanyal

Reviewer #2: No

Figure Files:

Data Requirements:

Reproducibility:

References:

---

## [Editor Report · Decision Letter 1]

4 Mar 2024

Dear Ms van Huizen,

We are pleased to inform you that your manuscript 'Deubiquitinating activity of SARS-CoV-2 papain-like protease does not influence virus replication or innate immune responses *in vivo*' has been provisionally accepted for publication in PLOS Pathogens.

Best regards,

Matthew B. Frieman

Guest Editor

PLOS Pathogens

Sonja Best

Section Editor

PLOS Pathogens

Michael Malim

Editor-in-Chief

PLOS Pathogens

orcid.org/0000-0002-7699-2064

We thank you for your resubmission. It answers all of the reviewers concerns. I thank you for the manuscript.
---

## [Editor Report · Acceptance letter]

8 Mar 2024

Dear Ms van Huizen,

We are delighted to inform you that your manuscript, "Deubiquitinating activity of SARS-CoV-2 papain-like protease does not influence virus replication or innate immune responses *in vivo*," has been formally accepted for publication in PLOS Pathogens.

Best regards,

Michael Malim

Editor-in-Chief

PLOS Pathogens

orcid.org/0000-0002-7699-2064